# Genomic epidemiology of syphilis reveals independent emergence of macrolide resistance across multiple circulating lineages

Mathew A. Beale [1], Michael Marks [2,3], Sharon K. Sahi[4], Lauren C. Tantalo[4], Achyuta V. Nori[5], Patrick French[6], Sheila A. Lukehart[7], Christina M. Marra[4] & Nicholas R. Thomson[1,8]

Syphilis is a sexually transmitted infection caused by *Treponema pallidum* subspecies *pallidum* and may lead to severe complications. Recent years have seen striking increases in syphilis in many countries. Previous analyses have suggested one lineage of syphilis, SS14, may have expanded recently, indicating emergence of a single pandemic azithromycin-resistant cluster. Here we use direct sequencing of *T. pallidum* combined with phylogenomic analyses to show that both SS14- and Nichols-lineages are simultaneously circulating in clinically relevant populations in multiple countries. We correlate the appearance of genotypic macrolide resistance with multiple independently evolved SS14 sub-lineages and show that genotypically resistant and sensitive sub-lineages are spreading contemporaneously. These findings inform our understanding of the current syphilis epidemic by demonstrating how macrolide resistance evolves in *Treponema* subspecies and provide a warning on broader issues of antimicrobial resistance.

[1] Parasites and Microbes, Wellcome Sanger Institute, Wellcome Genome Campus, Hinxton, Cambridgeshire, UK. [2] Clinical Research Department, Faculty of Infectious and Tropical Diseases, London School of Hygiene & Tropical Medicine, London, UK. [3] Hospital for Tropical Diseases, London, UK. [4] Department of Neurology, University of Washington, Seattle, WA 98195, USA. [5] Guy's & St Thomas' NHS Foundation Trust, London, UK. [6] The Mortimer Market Centre CNWL, Camden Provider Services, London, UK. [7] Departments of Medicine and Global Health, University of Washington, Seattle, WA 98195, USA. [8] Department of Pathogen Molecular Biology, Faculty of Infectious and Tropical Diseases, London School of Hygiene & Tropical Medicine, London, UK. Correspondence and requests for materials should be addressed to M.A.B. (email: mathew.beale@sanger.ac.uk) or to N.R.T. (email: nrt@sanger.ac.uk)

Syphilis is a centuries-old, predominantly sexually transmitted infection (STI) caused by the bacterium *Treponema pallidum* subspecies *pallidum* (TPA). If untreated, syphilis causes a multi-system disease that can progress to severe cardiovascular and neurological involvement, which can be potentially fatal. Syphilis caused a pandemic wave that swept across Renaissance Europe over 500 years ago, and remained a problem until the introduction of antibiotics in the post-World War II era[1]. Despite effective treatment with benzathine benzylpenicillin G (BPG), syphilis transmission levels fluctuated but persisted throughout the 20th century, until the AIDS crisis of the 1980s and 1990s, where changes in sexual behaviour (and possibly AIDS-related mortality), led to an overall decline in incidence in many western countries and populations[2,3].

Recent years have seen a sharp increase in syphilis cases in many high-income countries, predominantly within sexual networks of men who have sex with men (MSM)[4,5]. In the United Kingdom there was a 20% increase in reported new diagnoses between 2016 and 2017, and a 148% increase since 2008[6]. Similar trends have been reported in other countries[4,7]. The reasons for this increase are complex and multifactorial, incorporating changing behavioural patterns mediated by cultural, societal and technological changes in our modern world[8], resulting in a perfect epidemiological storm. It is also possible that there are bacterial changes either driving the current rise in syphilis incidence, or occurring as a consequence of this increase. However, current knowledge of TPA is limited, largely because the bacterium was, until recently, intransigent to in vitro culture[9]. Most current understanding of TPA biology therefore comes from related species or from TPA cultured in the in vivo rabbit testicular model[10]. Genomic analysis has also been limited due to low levels of TPA pathogen load in patients and difficulty in readily isolating new strains. Sequencing must be performed directly on clinical specimens or after passage through rabbits, leading to substantial bottlenecks in genomic data generation. Recent advances have enabled target enrichment of pathogen reads directly from clinical or cultured specimens[11,12], and this was recently employed separately by different groups, including our own, to sequence TPA and other *T. pallidum* subspecies directly from patient samples[13–15].

The availability of increasing numbers of genomic sequences enabled the first description of the global TPA population structure using 31 near genome-length TPA sequences, along with a small number derived from closely related species[14]. The authors described two lineages within TPA; a Nichols-lineage found almost exclusively in North American sequences exhibiting substantial nucleotide diversity, and a geographically widespread but genetically homogeneous SS14-lineage, confirming previous analyses using multi-locus sequence typing[16]. Of these two lineages, they found that 68% of tested TPA genomes belonged to the SS14-lineage, and further analysis using a larger dataset of 1354 single-locus molecular types (comprising 623 samples from South East Asia, 241 from the United States, 392 from Europe and a small number of other locations) also supported this view (94% SS14-lineage).

Although penicillin resistance has never been reported in syphilis, increasing levels of genotypic resistance, and clinical treatment failure, to macrolides such as azithromycin have been reported[17,18], conferred by either one of two single nucleotide polymorphisms (SNPs) in the 23S ribosomal sequence (A2058G and A2059G). Previous molecular typing studies indicated that differential patterns of macrolide resistance correlate with geographical location[17–19] and molecular type[18,20–22], and Arora et al. reported that 90% of whole-genome sequenced SS14-lineage genomes and 25% of Nichols-lineage genomes contained SNPs conferring macrolide resistance; furthermore, it has been suggested that SS14-lineage may represent a single pandemic azithromycin-resistant cluster[14] perhaps related to azithromycin selective pressure[14,16].

In this study, we use direct whole-genome sequencing on 73 TPA samples from the US and Europe, and combine these data with 49 publicly available genomes. We test the hypothesis that, as with other pathogens[23], expansion of the SS14-lineage might have been driven by selection of macrolide resistance. We use phylogenetic analysis to delineate sub-lineages within the both the SS14- and Nichols-lineages, providing an evolutionary framework for the emergence of macrolide resistance in treponemas, and showing striking patterns of the emergence and fixation of macrolide resistance SNPs that indicate independent evolution and proliferation of resistance alleles. These findings have implications for the potential of the WHO Yaws eradication campaign to drive further development of macrolide resistance in both TPA and in the closely related *Treponema pallidum* subspecies *pertenue* (TPP)[24,25].

## Results

**Clinical genomes are mostly but not exclusively SS14-lineage.**
We sequenced eight genomes directly from clinical swabs collected in 2016 from patients in the United Kingdom and 60 isolate genomes from low rabbit passage samples (no more than two passages from the original patient sample; henceforth referred to as recently clinically derived) originally collected from patients between 2001 and 2011 in the United States. We also resequenced three clonally derived laboratory strains from the United States that have been previously sequenced (Nichols Houston E, Nichols Houston J, Nichols Houston O) but remain unpublished, and two strains for which the sequencing reads were not publicly available (Chicago[26], Seattle 81-4[27]). We combined our data with 49 high-quality genomes published previously[13,14,28–33], 41 of which were recently derived from clinical patients, yielding a dataset of 122 genomes (109 with limited passage from clinical patients). Combined, our sample set included 72 genomes from the United States (predominantly Seattle), 8 from the United Kingdom (exclusively London), 9 from China (predominantly Shanghai), 23 from Portugal (exclusively Lisbon), and a small number from other countries, all collected between 1912 and 2016 (Supplementary Data 1).

After removal of recombinant and repetitive sites (both by selective mapping and screening—see Methods and Supplementary Data 2), we performed phylogenomic analyses, using maximum likelihood and Bayesian methods to define lineages. In agreement with previous studies[14], we show the presence of two dominant lineages in our dataset (previously denoted SS14 and Nichols; Fig. 1) that are separated by > 70 non-recombining single nucleotide polymorphisms (SNPs). Of the 122 total samples included in this study, 105 (86%) belonged to the SS14-lineage, whilst of the 109 recently clinically derived samples included, 103 (94%) were from SS14-lineage. In contrast, only six Nichols-lineage samples were recently clinically derived, and most (11/17) Nichols-lineage genomes examined were historically passaged isolates, including those derived from the original Nichols strain isolated in 1912 and disseminated to different North American laboratories; some Nichols-lineage genomes represent clones of the parent strain derived in vivo. However, although we observed a strong bias towards clinically derived SS14-lineage samples in this dataset, not all recent clinical strains were of the SS14-lineage; six recent clinical samples belonged to the Nichols-lineage, three (of eight sequenced) clinical samples from the United Kingdom in 2016, one collected in the Netherlands in 2013, and two from the United States in 2004, indicating that transmission of this lineage is ongoing and potentially more widespread than previously thought.

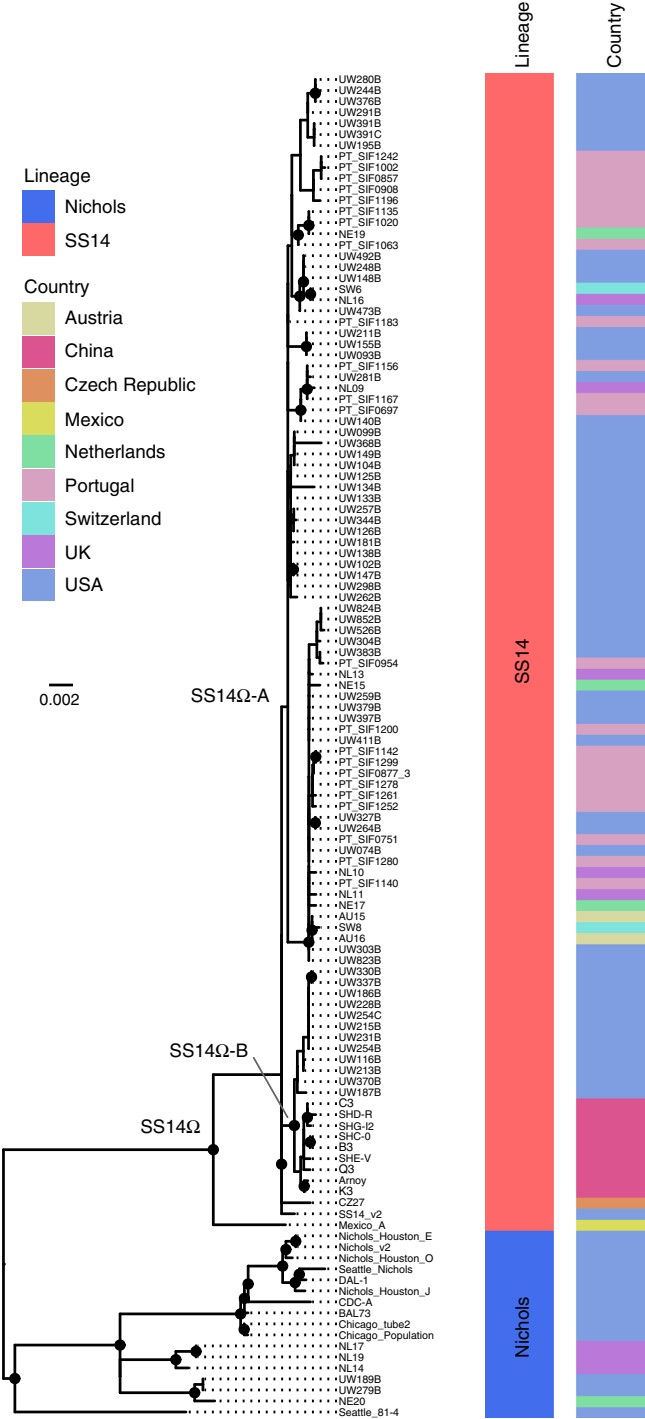

**Fig. 1** Maximum likelihood phylogeny of 122 high-quality *T. pallidum* subspecies *pallidum* genomes, showing lineage and country of origin. Tree includes sequences from recently clinically derived samples and those with more extensive laboratory passage. Ultra-Fast bootstrap values > = 95% are labelled with black nodes points. Branches are scaled by mean nucleotide substitutions/site

Bayesian phylogenetic reconstruction was used to date the time to most recent common ancestor (TMRCA) for the different TPA lineages. However, temporal analysis of heavily passaged laboratory strains (such as those derived from the original Nichols isolate) is problematic because the true mutational age may be unknown, meaning coalescent date is difficult to infer; we therefore removed extensively passaged strains or strains with no

record of their passage history from this particular part of the analysis. This included the removal of the Nichols and SS14 reference strains, as well as the Mexico A strain that delineated the SS14Ω lineage described previously[14]. Root-to-tip regression analysis of the remaining 109 recently clinically derived genomes indicated that TPA possesses a clock-like signal (Supplementary Figure 1), and we performed Bayesian phylogenetic reconstruction and tip date analysis using BEAST[34] under a Strict Constant model, inferring a median molecular clock rate of $1.78 \times 10^{-7}$ (95% Highest Posterior Density (HPD) $1.15 \times 10^{-7} - 2.44 \times 10^{-7}$), or 0.20 sites/genome/year (meaning we would expect TPA genomes on average to accumulate one fixed SNP every 5 years). We inferred a temporal timeline for the tree, and our analysis broadly supported previous estimates[14] dating the separation of Nichols- and SS14- lineages between the 17th and 18th Centuries (median date 1662, 95% HPD 1517–1791; Fig. 2a).

**SS14-lineage shows a polyphyletic structure.** Within the SS14-lineage, the high number of full-length sequences enabled fine-scale description of phylogenetic sub-structure. In particular, we show partitioning of the SS14Ω centroid cluster previously defined[14] into two lineages; one composed of European and North American derived samples (SS14Ω-A), and another of Chinese and North American derived samples (SS14Ω-B) (Fig. 1). While the American and European samples belonging to the former SS14Ω-A lineage are geospatially admixed, those of the latter SS14Ω-B lineage can be further separated between Chinese and North American samples. These partitions were well supported in our maximum likelihood (Fig. 1) and Bayesian phylogenies (Fig. 2a), as indicated by black node points. We used the rPinecone package[35] to formally classify these sub-lineages based on a defined root-to-tip SNP distance, identifying eight sub-lineages (one of which we further subdivided into sub-lineages 1A and 1B to aid analysis based on temporal and geospatial divergence) within SS14-lineage that correlated well with the population structure described by the phylogeny (Fig. 2a, Supplementary Figure 2). Importantly, while some nodes close to the tips in our phylogeny are unsupported due to small numbers of differentiating SNPs, all sub-lineages defined by rPinecone are supported by >91% posterior support at the key nodes in our Bayesian phylogeny (Fig. 2a). We considered the possibility of geographical bias in our data, and whilst four sub-lineages are composed entirely of samples from Seattle (sub-lineages 1A, 3, 4) or China (sub-lineage 1B), four others (sub-lineages 2, 5, 7, 8) are geospatially admixed, sharing samples from both the United States and Europe, with the last (sub-lineage 6) composed of samples from two European countries, Portugal and the Netherlands (Supplementary Figures 2 and 3, Supplementary Data 1).

**Macrolide resistance has evolved independently within SS14.** The molecular basis for macrolide resistance has been well documented in *T. pallidum*, and is mediated by point mutations in the 23S ribosomal RNA gene at nucleotide positions 2058 and 2059[36–38]. The A2058G variant was first identified in *T. pallidum* Street Strain-14 (the prototype sample for the SS14-lineage), isolated as long ago as 1977[36], yet resistance has not previously been analysed in context with a detailed whole-genome phylogeny. We initially used ARIBA[39] to perform localised assembly and variant calling of treponema-specific 23S ribosomal sequences from all genomes, and these data were used to infer the presence of both A2058G and A2059G 23S variants that confer macrolide resistance[36]. *T. pallidum* possesses two copies of the 23S ribosomal RNA gene, yet previous analyses of *T. pallidum* have not identified heterozygosity between these two copies (although heterozygosity was described for the distantly related *T.*

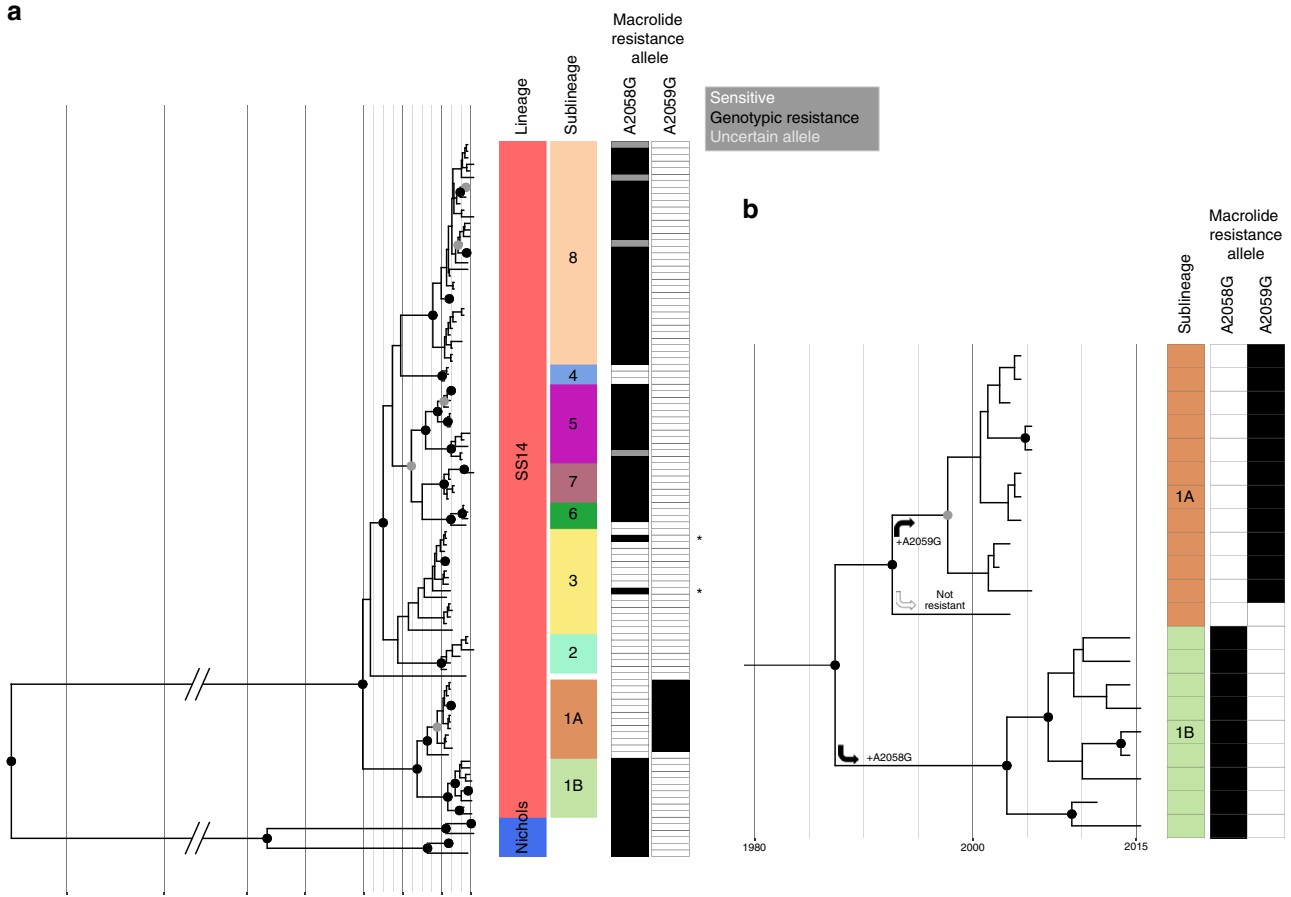

**Fig. 2** Bayesian maximum credibility phylogeny shows expansion of discrete sub-lineages within SS14-lineage, with independent evolution of macrolide resistance. **a** Time-scaled phylogeny of all recently clinically derived genomes. Coloured tracks indicate lineage, sub-lineage, and presence of macrolide resistance conferring 23S rRNA SNPs (black = present, white = absent, grey = uncertain—alleles in some samples could not be definitively classified either because they appeared mixed or because of low numbers of reads at each site). Node points are shaded according to posterior support (black ≥ 96%, dark grey > 91%, light grey > 80%). *Sporadic (non-lineage associated) gain of resistance is highlighted in sub-lineage 3 (samples UW133B and UW262B). **b** Expanded view of sub-lineages, showing independent acquisition and fixation of macrolide resistance alleles

*denticola*[40])—where resistance alleles have been sequenced, they are homozygous between 23S copies, and it has been suggested a gene conversion unification mechanism may exist to facilitate this[18]. In our initial analysis of the 122 genomes, 83 showed evidence of genotypic resistance to macrolides, with 76 genomes showing > 95% read support for either the A2058G or A2059G variant. Since it is not possible to discriminate between short reads originating from either copy of 23S because they are perfect repeats, this suggests that both copies carry the same resistance mutation. Seven clinically derived genomes (one UK sample from this study, six described by Pinto and colleagues[13]) showed a mixed 23S allelic profile (replicating the result of a previous analysis for three of six samples[13]). All of these samples had > 179x read coverage for those sites, with only a fraction of reads (26–94%) possessing a resistance allele. In these cases it was not possible to clearly distinguish between a mixture of homozygous positive and negative bacteria in the same patient (either due to within-host evolution or coinfection with multiple strains) or heterozygous sequences from a single bacterium (different 23S rRNA alleles at each copy; heterozygosity in phase) (Fig. 2a).

In the course of our analysis, we determined that many of the samples with mixed alleles from publicly available datasets also contained contaminating reads from other species despite the use of sequence capture based enrichment approaches; many of these contaminating reads corresponded to genomic regions with

substantial homology to other bacterial species. In particular, the 23S is broadly conserved across bacteria, with the region immediately preceding the 2058/2059 locus sharing 100% k-mer identity with over 326,000 other bacterial datasets, and this can lead to problems for both assembly and mapping approaches to SNP calling in this region[41]. For example, Kraken[42] analysis showed that the read set for sample SW6 (accession SRR3268732), which in our initial analysis using ARIBA appeared to be negative for the A2058G SNP, actually contained 29% *Streptococcus galactiae* reads, most of which clustered in the 23S, where read depth reached over 70,000x for the region immediately preceding the 2058/2059 locus. This can lead to failures of both assembly and mapping approaches, since reads homologous to this region would have anomalous mate pairs. Careful manual examination of the reads showed that the SNP was indeed present, and we used stringent competitive mapping against comparable genomic regions from multiple species as described by Hadfield and colleagues[43], whereby only reads with appropriately mapping mates are considered, to deconvolute these mixtures throughout the dataset. Using this approach, we determined that SW6 was indeed genotypically resistant at A2058G, as well as demonstrating that the majority of mixed alleles detected were in fact the result of contaminating reads, and that the *Treponema*-derived reads were positive for A2058G. Where competitive mapping could not conclusively establish the

genotype of alleles, they are described as "uncertain" throughout this manuscript.

Within the 122 genomes included in this study, we observed that 71% (74/105) of SS14-lineage samples and 35% (6/17) of Nichols-lineage samples were homozygous for either A2058G or the A2059G 23S rRNA allele. In the SS14-lineage, samples possessed either the A2058G ($n = 63$) or the A2059G ($n = 11$) variant. In the Nichols-lineage, six possessed a resistance allele, of which all showed the A2058G variant, and all were from recent clinically derived samples.

To explore the emergence of macrolide resistance, we correlated the taxa in our time-scaled phylogeny with the presence of resistance alleles (Fig. 2a). We observed a strong correlation between our well supported sub-lineages and genotypic macrolide resistance or sensitivity, such that resistance appears to have evolved on multiple occasions in a stepwise manner (Fig. 2a). For example, Fig. 2b shows how the wild-type ancestor of sub-lineage 1B sequences evolved the A2058G between the late 1980s and late 1990s, contrasting with sub-lineage 1A sequences which did not gain A2058G, but subsequently and independently evolved the A2059G variant.

More broadly in the phylogeny, we observe that similar independent 23S rRNA mutations have occurred on at least six occasions (Fig. 2a), with separate sub-lineages characterised by either predominantly macrolide resistant or sensitive genotypes; six of the nine sub-lineages were predominantly resistant to macrolides, with three sub-lineages being predominantly sensitive. Several of the macrolide sensitive sub-lineages (in particular sub-lineage 3) contained similar numbers of samples as many of the resistant sub-lineages (Fig. 3), suggesting ongoing selection for macrolide resistance has not influenced the expansion of these sub-lineages.

To examine sub-lineage expansion in greater detail, we extracted sampling dates according to sub-lineage, and correlated these data with the predicted time to most recent common ancestor (TMRCA) (Fig. 3). Whilst all clinical sequences included in this analysis were sampled after 2000, our analysis indicates that the origins of most of the sub-lineages predated this time and likely arose between the late 1980s and early 2000s (Fig. 3).

Regarding whole sub-lineage associated resistance, we also observed more recent sporadic appearance of resistance mutations, with two separate A2058G variants detected in sub-lineage 3 (an otherwise macrolide sensitive sub-lineage) (Fig. 2).

**Penicillin binding proteins show homoplasic changes.** Previous analysis of Treponemal genomes have shown amino acid changes in penicillin binding proteins[13,28], but large scale analysis of penicillin gene variants has not previously been performed on a global dataset. We selected three penicillin binding protein genes[44,45] (*pbp1*, *pbp2*, *mrcA*) and a putative β-lactamase[46] (*Tp47*), and used competitive mapping to individual gene sequences to screen for novel variants across our full clinical dataset of 109 sequences, identifying 13 non-synonymous SNPs (Supplementary Figure 4). Two variants (*pbp1* P564L; *Tp47* S394R) were almost universally conserved in clinical SS14-lineage sequences, whilst three more (*mrcA* I487L; *mrcA* M625V; *mrcA* G708S) showed strong association with SS14 sub-lineages. As no phenotypic or clinical evidence exists for penicillin resistance or treatment failure in Treponemes, we considered such widely prevalent SNPs unlikely to be of clinical relevance or mediate a resistant phenotype.

Sun and colleagues[28] identified variants in *pbp2* (*pbp2* I415F/I415M) and *mrcA* (*mrcA* A506V) in eight genomes from China, and whilst we confirmed the presence of the *pbp2* variants, we did not find them in samples from other countries. However, in addition to the *mrcA* A506V variant described by Sun et al., we also identified a SNP affecting the same amino acid position (*mrcA* A506T), both in the recently clinically derived Nichols-lineage genomes as well as in recently clinically derived SS14 sub-lineage 1A (comprising samples from the United States). As with macrolide resistance, independent evolution of homoplasic variants affecting the same amino acid are suggestive of selection, and we used SIFT[47] to infer the likely functional impact of amino acid changes at *mrcA* A506 and *pbp2* I415. None of the four variants detected showed a significant result (no SIFT scores were below the threshold for functional effect of 0.05), leaving the clinical relevance of these SNPs unclear.

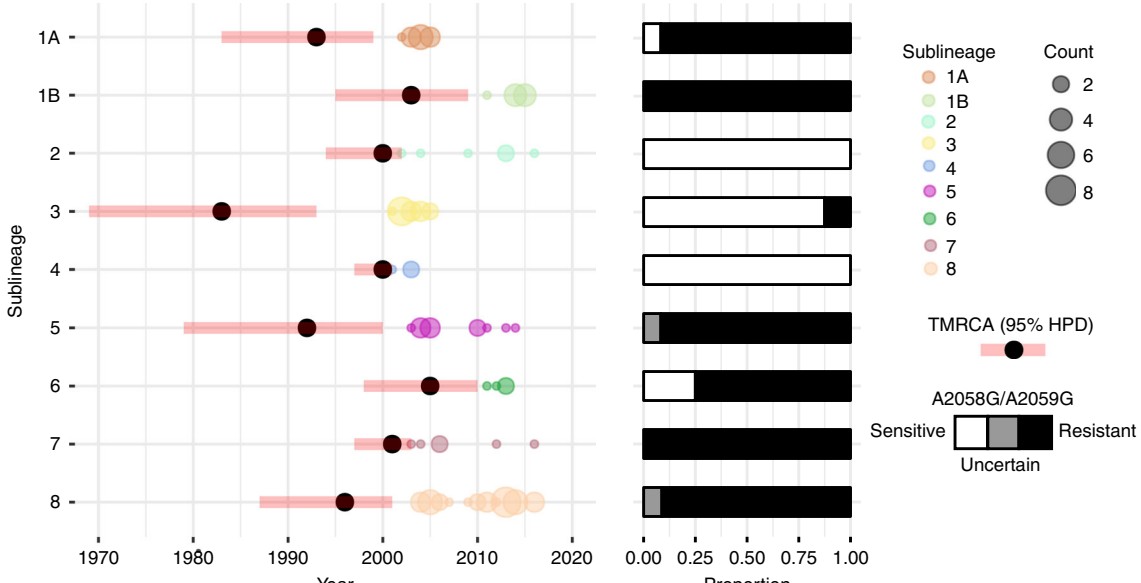

**Fig. 3** Macrolide resistant and sensitive SS14 sub-lineages evolved independently prior to 2006 and expanded equally regardless of resistance genotype. This figure shows sample collection dates grouped by sub-lineage, with size of coloured circle proportional to number of sequences, and showing predicted time to most recent common ancestor (TMRCA; black circle) with 95% highest posterior density (HPD; red bars), and proportion of genotypically macrolide resistant (black), sensitive (white) and uncertain (grey) samples

## Discussion

Compiling the largest *Treponema pallidum* subspecies *pallidum* sequence collection to date, we show that the majority of the contemporary samples sequenced here were from the SS14-lineage, consistent with other reports[14]. However, three of eight recent UK genomes (38%) belonged to the Nichols-lineage, showing that these two lineages are still circulating concurrently, and that the prevalence of Nichols-lineage strains may vary by sampling population. Moreover, linking the phylogenetic and geospatial data of our samples shows that a number of SS14 sub-lineages (2, 5, 7, 8) are comprised of both European and North American samples (Supplementary Figure 2, Supplementary Figure 3), suggesting that we have captured key lineages circulating in these regions and that our data are not overly biased from localised sampling. Notwithstanding the above, further studies will be needed to more accurately characterise the genomic diversity and prevalence of lineages within different populations on a global scale.

We were able to reconstruct a time-scaled phylogeny using only recently clinically derived samples, and the increase in whole-genome sequence numbers combined with the removal of heavily passaged samples contrasts with previous approaches[14,28]. Arora et al. reported a mean evolutionary rate of $6.6 \times 10^{-7}$ substitutions/site/year for *T. pallidum*[14], comparable with rapidly dividing, free-living bacterial pathogens with environmental life cycles such as *Vibrio cholerae* ($6.1 \times 10^{-7}$)[48] and *Shigella sonnei* ($6.0 \times 10^{-7}$). However, *T. pallidum* is a slow growing host-restricted pathogen with substantial periods of latency, and as such we would expect a molecular clock rate more similar to other slow growing organisms such as *Chlamydia trachomatis* ($2.15 \times 10^{-7}$)[43] or *Mycobacterium tuberculosis* ($1 \times 10^{-7}$)[49]. Our inferred rate for TPA ($1.78 \times 10^{-7}$) is consistent with this expectation, as well as with other observations that suggest *T. pallidum* has a low evolutionary rate[50].

Within the SS14-lineage, we defined nine well supported sub-lineages that all diverged from their most recent common ancestors prior to 2006, with the earliest (sub-lineage 3) potentially emerging during the 1980s. We observed clear associations between these nine sub-lineages and the presence of macrolide resistance conferring SNPs, with each sub-lineage dominated by either macrolide resistant (six sub-lineages) or macrolide sensitive (three sub-lineages) samples. Such observations, previously documented to a limited extent within a single country[13] but now extended to a broader sample, are not consistent with a hypothesis of an ancestrally resistant SS14-lineage driven to high frequency in the population due to a fitness advantage conferred by macrolide resistance, where we would expect to see expansion of a single resistant lineage. Rather, we see evidence of multiple sub-lineages independently evolving macrolide resistance alleles, as a likely consequence of intermittent selective pressure from macrolide treatment, consistent with molecular typing data from Seattle[20]. Phylogenetic reconstruction shows de novo evolution of macrolide resistance in syphilis is not a rare event, and further-more, when resistance evolves in a lineage, it persists in descendants, resulting in transmission from person to person. That the variants appear stable within lineages, with no examples in the phylogeny that might represent reversion to a wild-type state, suggests that either there is no strong fitness cost associated with possessing these macrolide resistance mutations or that novel compensatory variants co-evolve to mitigate any fitness cost, enabling variant stability; further sampling would be necessary in the future to address this.

Although our data strongly suggest that global expansion of the SS14-lineage is not contingent on macrolide resistance, the global increases seen in macrolide resistant syphilis[18], as well as the number of resistant lineages emerging in our data, are a cause for concern. Macrolides, such as azithromycin, are not considered appropriate treatment for syphilis, with WHO and US guidelines recommending treatment with BPG[51,52], with doxycycline recommended as a secondary treatment option. In contrast, WHO now recommends azithromycin rather than BPG as the treatment of choice for mass drug administration in the eradication of yaws[24], caused by the closely related *T. pallidum* subspecies *pertenue*. Macrolide resistance has recently been described in yaws[25] and our data suggest that further independent evolution of azithromycin resistance is highly likely, which would have significant implications for yaws eradication efforts. Worryingly, applying azithromycin-based mass drug administration to populations infected with both TPP and TPA could promote resistance in both species.

Several factors are likely to have driven the repeated evolution and subsequent expansion of macrolide resistant lineages of TPA. Use of azithromycin, or other macrolides, for other indications is likely to have played a significant role[53]. Azithromycin entered global markets between 1988 and 1991 (marketed by Pfizer as Zithromax), and became one of the most widely used antibiotics in the United States for a wide variety of indications[54], including the treatment of respiratory tract infections and for the treatment of other sexually transmitted infections. In many cases, the dose used for treatment of these indications is lower than the recommended dose for the treatment of syphilis. Azithromycin and clarithromycin were also widely used prophylactically amongst individuals living with HIV prior to the widespread availability of combined anti-retroviral therapy. Off-target macrolide exposure is of particular concern because azithromycin has a long half-life[55] and may persist at sub-therapeutic concentrations in patients. Widespread use of macrolides for this broad range of indications might therefore have contributed to sub-therapeutic exposure of patients with incubating or early syphilis and ultimately selection of resistance[53].

The recent increase in incidence of syphilis in high-income countries likely reflects changes in sexual behaviour[8]. The fact that we observe the expansion of both genotypically resistant and sensitive lineages highlights particular treatment issues. There have been significant global shortages of BPG[56] in low, middle, and high-income nations, including the United States, with the global supply of BPG dependent on just three manufacturers of the active ingredient[56]. Pharmaceutical production of sterile, injectable β-lactam-derived antimicrobials such as BPG is costly, yet as an older off-patent medication with declining demand in the face of growing antimicrobial resistance (AMR) in other organisms, the financial rewards for production are low[56]. This may be compounded by a misperception that BPG is an outdated drug that could be replaced by newer, more effective drugs[56]. In circumstances where azithromycin is used instead of doxycycline, a shortage of BPG leads inevitably to inadequate treatment of early infectious syphilis and contributes to ongoing, unchecked transmission. In China for example, studies have reported that despite high rates of macrolide resistance[19], clinicians have inappropriately been resorting to macrolide treatment due to ongoing BPG shortages[57]. Thus although a well-established, highly effective treatment for syphilis (BPG) has been available since the mid-1950s, shortages in the present era contribute to suboptimal treatment strategies and continued use of drugs with a known resistance problem.

Phenotypic or clinical penicillin resistance has never been observed in syphilis in more than 60 years of clinical use. In our analysis, we detected a number of variants affecting penicillin binding proteins, although given that penicillin remains an effective treatment, many were so widely distributed throughout the phylogeny as to be implausible for causing resistance.

The epidemic of syphilis and the widespread problems of azithromycin resistance and BPG shortage require a multi-faceted response. This includes new strategies for treatment and reduction of transmission, finding ways to improve the security of the BPG supply chain, and strengthening molecular surveillance for antimicrobial resistance in *T. pallidum*[58]. Many authors have discussed the importance of rethinking the economics of anti-microbial development pipelines to ensure we are still able to treat infections[23,59]. In syphilis, we must rethink how we can protect the continued production of existing highly efficacious penicillins in the face of increasing antimicrobial resistance rendering them ineffective for other organisms, especially in the light of the recent increases in syphilis incidence in Europe, North America, and Asia.

## Methods

**Samples.** UK samples consisted of residual DNA, extracted from clinical swabs using a QIAsymphony (Qiagen) from routine diagnostic samples obtained from patients presenting with clinical evidence of syphilis at the Mortimer Market Centre, London. Use of the UK samples was approved by the NHS Research Ethics Committee (IRAS Project ID 195816). US samples from Seattle were collected from individuals enrolled in a study of cerebrospinal fluid abnormalities in patients with syphilis (a mixture of 21 patients with evidence of CSF infection and 38 patients without), with ethical approval at the University of Washington (UW IRB # STUDY00003216). All relevant ethical regulations for work with human participants were followed, and informed consent was obtained for sampling and research.

For US samples, 2.4–3.0 ml participant blood or cerebrospinal fluid was inoculated intratesticularly into male New Zealand white rabbits (~3 kg) testes under sedation. The rabbits were housed at 18–20 °C and fed antibiotic-free food. Each animal was inspected at regular intervals for signs of orchitis (testicular swelling and hardness), which occurs variably between 10 and 50 days depending on the strain. When infection was suspected (confirmed by positive darkfield examination of testicular aspirate or seroconversion), rabbits were euthanised, the popliteal lymph nodes and testes removed, then minced and resuspended (50% normal rabbit serum, 50% saline) and inoculated into a naive rabbit within 30–60 min of collection. *T. pallidum* suspensions collected after the second round of passage were used for DNA extraction and sequencing. This well-established passage method has been described elsewhere in detail; for specific protocols see Lukehart and Marra[60]. Historical strains were propagated in rabbits and harvested from infected testes. Animal care was provided in full accordance with established guidelines, and experimental procedures were conducted under protocols approved in advance by the University of Washington Institutional Animal Care and Use Committee.

*T. pallidum* suspensions from rabbits were treated using a lysis buffer (10 mM Tris pH 8.0, 0.1 M EDTA pH 8.0, 0.5% SDS), freeze-thaw, and extraction using QIAamp Mini kit (Qiagen) according to the manufacturer's instructions; in select cases the proteinase K incubation was extended overnight to improve DNA yield. In some cases, samples had been used for previous molecular tests, and in these cases residual DNA extracts were used for sequencing. Treponemal DNA was quantified using a qPCR targeting the Tp47 (TPANIC_0574) gene that is conserved across all known members of the *T. pallidum* cluster, and compared with a standard curve derived from a plasmid containing the PCR amplicon. Samples with a concentration > 2000 genome copies/μl were selected for sequencing; borderline samples with high volume and a pathogen load over 500 genome copies/μl were concentrated using a vacuum centrifuge. Samples were arranged in groups of 20 according to similar (within $2C_T$) treponemal load, with high concentration outlier samples diluted as necessary. We added 4 μl pooled commercial human gDNA (Promega) to all samples to ensure total gDNA > 1 μg/35 μl, sufficient for library prep.

**Sequencing.** Genomic DNA was sheared to 100–400 bp (average size distribution ~ 150 bp) by ultrasonication (LE220, Covaris Inc), followed by library preparation (NEBNext Ultra II DNA Library prep Kit, New England Biolabs), adaptor ligation and index barcoding (Sanger 168 tag set). Index tagged samples were amplified (six cycles of PCR, KAPA HiFi kit, KAPA Biosystems), quantified (Accuclear dsDNA Quantitation Solution, Biotium), then pooled in the preassigned groups of 20 to generate equimolar total DNA pools. Five hundred nanograms of pooled material was then hybridised using 120-mer RNA baits (SureSelect Target enrichment system, Agilent Technologies) designed against published RefSeq examples of *T. pallidum* and *T. paraluiscuniculi* (Bait design ELID ID 0616571)[15]. Libraries were subjected to 125 bp paired-end sequencing on Illumina HiSeq 2500 with version 4 chemistry according to established protocols. This sequencing method has been previously described[15], and differs from that of others[13,14], in that samples are barcoded and pooled before enrichment thus enabling multiplexing of hybridisation reactions. Raw sequencing reads were deposited at the European Nucleotide Archive (ENA) under project PRJEB20795; all accessions used in this project are listed in Supplementary Data 1.

**Sequence analysis and phylogenetics.** Treponemal sequencing reads were pre-filtered using a Kraken[42] v0.10.6 database containing all bacterial and archaeal nucleotide sequences in RefSeq, plus mouse and human, to identify and extract those reads with homology to Treponema species, followed by adaptor trimming using Trimmomatic[61] v0.33. To reduce bias due to variable read depth, as well as make analysis computationally tractable, for samples with high read counts we used seqtk v1.0 (available at https://github.com/lh3/seqtk) to randomly down-sample the binned and trimmed reads to 2,500,000 unique treponemal read pairs. For publicly available genomes, raw sequencing reads were downloaded from SRA and subjected to the same binning and down-sampling pipeline. For eight public genomes (see Supplementary Data 1) raw sequencing reads were not available; for these we simulated 125 bp PE perfect reads from the RefSeq closed genomes using Fastaq (available at https://github.com/sanger-pathogens/Fastaq).

For phylogenetic analysis, we used a reference mapping approach with a custom version of the SS14 v2 reference sequence (NC_021508.1) from which we first masked 14 highly repetitive or recombinogenic genes (12 repetitive Tpr genes A-L, arp and TPANIC_0470) using bedtools[62] v2.17.0 maskfasta. We mapped prefiltered sequencing reads to the reference using BWA mem[63] v0.7.17 (MapQ ≥ 20), followed by indel realignment using GATK[64] v3.4-46 IndelRealigner, deduplication with Picard MarkDuplicates v1.127 (available at http://broadinstitute.github.io/picard/), and variant calling and consensus pseudosequence generation using samtools v1.2[65] and bcftools v1.2, requiring a minimum of three supporting reads per strand and eight in total to call a variant, and a variant frequency/mapping quality cut-off of 0.8; sites not meeting these criteria were masked to 'N' in the pseudosequence. Importantly, reads mapping to multiple positions were marked and excluded from SNP calling, meaning repeated regions such as the duplicated 23S genes was not included in the multiple sequence alignment used to derive the phylogenies.

Multiple sequence alignments were screened for evidence of recombination using Gubbins[66], generating recombination-masked full-genome length and SNP-only alignments. Maximum likelihood phylogenies were calculated on SNP-only alignments using IQ-Tree v1.6.3[67], correcting for missing constant sites using the built-in ascertainment bias correction[68], allowing the built-in model testing[69] to determine a K3P (three substitution types model and equal base frequencies) substitution model[70] with a FreeRate model of heterogeneity[71] assuming three categories, and performing 1000 UltraFast Bootstraps[72].

To determine SS14 sub-lineages, we recalculated a maximum likelihood tree as described above for SS14-clade sequences only (using the Mexico A strain, NC_018722.1 as outgroup), before performing a joint ancestral reconstruction[73] of SNPs on the tree branches using pyjar (available at https://github.com/simonrharris/pyjar). We then used the rPinecone package[35] (available at https://github.com/alexwailan/rpinecone), which applies a root-to-tip approach to defining clusters based on SNP distance relative to ancestral nodes. We used a cluster threshold of 10 SNPs, which proved optimal for describing the underlying phylogenetic structure of the tree, and yielded eight sub-lineages. Within sub-lineage 1, we found that pinecone clusters did not accurately represent the phylogeny, despite a clear phylogenetic separation, with one group of sequences from China associated with the A2058G allele, and the other group from the United States associated with the A2059G allele. We further manually clustered these sequences according to their shared ancestral nodes, naming them sub-lineages 1 A and 1B.

We evaluated our maximum likelihood phylogeny for evidence of temporal signal using TempEst[74] v1.5, and this showed a correlation of 0.40 and $R^2$ of 0.16 for the whole tree (Supplementary Figure 1), whilst the SS14-lineage -only alignment showed a correlation of 0.66 and $R^2$ of 0.44; this indicated that there was sufficient evidence for temporal signal and we proceeded to BEAST analysis. BEAST[34] v1.8.2 was initially run on a recombination-masked SNP-only alignment containing 276 variable sites, applying a correction for invariant sites using the constantPatterns argument, in triplicate using an Uncorrelated Relaxed Clock model, assuming constant population size, lognormal population distribution, GTR substitution model, diffuse gamma distribution prior (shape 0.001, scale 1000), with a burnin of 10 million cycles followed by 100 million MCMC cycles. All MCMC chains converged, and on inspection of the marginal distribution of ucld. stdev we could not reject a Strict Clock. We therefore repeated the analysis using a Strict Clock model, using the same models and priors and assuming a starting molecular clock rate of $3.6 \times 10^{-4}$ as described by others[14]. We used the marginal likelihood estimates from the triplicate BEAST runs as input to Path Sampling and Stepping Stone Sampling analyses[75,76] and determined that the Strict Constant model was optimal for this dataset. To further confirm the temporal signal in our tree, we used the TIPDATINGBEAST package[77] in R[78] to resample tip-dates from our alignment, generating 20 new datasets with randomly assigned dates—BEAST analysis using the same Strict Clock prior conditions found no evidence of temporal signal in these replicates, indicating that the signal in our tree was not found by chance (Supplementary Figure 5).

Macrolide resistant SNPs were initially inferred using ARIBA[39], which performs localised assembly and mapping in comparison with a custom reference database containing 23S sequences from Nichols (NR_076156.1) and SS14 reference strains (NR_076531.1). Detailed inspection of ARIBA results revealed inconsistencies relating to both contamination of publicly available datasets with other species and low coverage. We therefore repeated the analysis using a stringent competitive mapping approach[43]. Reads were simultaneously mapped to a region spanning 16S, 23S and 5S from both the TPA Nichols genome and the most common contaminant in the public

datasets (*Streptococcus disgalactiae*; NC_019042.1). Reads with anomalous mapping, mapQ < 30, or for which only one of a read pair mapped were discarded, and TPA variants were called with bcftools v1.2, requiring a minimum of two supporting reads per strand, five in total to support a variant, and twelve at each position, as well as requiring alleles not pass a Chi Square test ($p < 0.01$) for the presence of strand bias, map quality bias, base quality bias and tail bias (available from bcftools in the PV4 option) using code available at https://github.com/matbeale/Global_Syphilis_Phylo_2019. Sites not meeting these criteria, or those showing a mixed allele profile (10–90% of reads) were marked as uncertain in all analyses.

Non-synonymous penicillin binding protein gene variants in *pbp1* (TPANIC_0500), *pbp2* (TPANIC_0760), *mrcA* (TPANIC_0705) and *Tp47* (TPANIC_0574) were inferred using the same stringent mapping approach described above, using gene sequence references corresponding to the nucleotide sequences from the Nichols genome (NC_021490.2). Functional inference of amino acid changes was performed by inputting the gene amino acid sequences into the SIFT[47] server (available at http://sift.bii.a-star.edu.sg). The treponema-specific *Tp47* protein (TPANIC_0574) did not return a result from SIFT due to insufficient sequences being found during the PSIBLAST step.

Processing of data, and all statistical analysis was performed in R[78] v3.4.1, primarily using the phytools and ape packages. Phylogenies were plotted using ggtree[79], and figures were produced using ggplot2[80].

**Reporting summary**. Further information on research design is available in the Nature Research Reporting Summary linked to this article.

## Data availability
Raw sequencing reads for all novel sequences were deposited at the European Nucleotide Archive (ENA) under project PRJEB20795. All accessions (both novel and previously published) used in this project are listed in Supplementary Data 1, along with all metadata used for analysis in Figs. 1, 2 and 3, and Supplementary Figures 1, 2, 3, 4 and 5. The maximum likelihood phylogeny and metadata from this study are also available for interactive exploration in a Microreact project [https://microreact.org] at https://microreact.org/project/rJ6mBc-rm/9c1fdf1c.

## Code availability
All code and source files for downstream analysis are available at https://doi.org/10.6084/m9.figshare.7688093.

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

## Acknowledgements

The authors thank the sequencing team at Wellcome Sanger Institute; S Harris for access to local scripts; A Wailan for discussions around rPinecone; D Domman for helpful advice and discussion during analysis; M Fookes for initial pilot work and bait design for SureSelect; B Molini and C Godornes at University of Washington for provision of strains; staff at University College London Hospital and clinical and sexual health staff at the Mortimer Market Clinic, London for UK sample provision. M.M. was supported by a Wellcome Trust Clinical Research Fellowship (102807). Strains from the University of Washington were supported by NIH grants R01 NS34235 to C.M.M., and R01 AI 34616 and R01 AI 42143 to S.A.L. M.A.B. and N.R.T. are supported by Wellcome funding to the Sanger Institute.

## Author contributions

Conceived and designed the study: N.R.T., S.A.L., M.A.B., A.V.N. and M.M. Collected and collated samples: S.A.L., C.M.M., M.M., A.V.N. and P.F. Performed the laboratory work: M.A.B., S.K.S. and L.C.T. Analysed the data: M.A.B. Wrote the initial draft of the paper: M.A.B. All authors viewed and contributed to the final paper.

## Additional information

**Competing interests:** The authors declare no competing interests.

