## [Peer Review File · Nature Communications]

Reviewers' Comments:

Reviewer #1:

Remarks to the Author:

The manuscript „Genomic epidemiology of syphilis reveals independent emergence of macrolide resistance across multiple circulating lineages“ by Beale and colleagues describes the whole genome sequencing of 73 TPA samples from US and Europe and whole genome analysis of obtained data together with 48 publicly available TPA genomes.

This reviewer appreciates the amount of work done by the authors. However, although the sequencing of 73 TPA samples brings new data, conclusions and methodology are not novel. The sample enrichment using nucleic acid baits was described before (Arora et al. 2016, Pinto et al. 2016, Marks et al. 2018, Knauf et al. 2018) and emergence of macrolide resistance in different samples was described in a number of TPA typing papers (Chen et al. 2013, Grillova et al. 2014, Gallo Vaulet et al. 2017, Smajs et al. 2015).

The reviewer wants to stress that there is a number of statements throughout the paper, which are controversial and these include:

1/ lines 39-43, “These findings demonstrate that macrolide resistance has independently evolved multiple times in *T. pallidum*, that once evolved it becomes fixed in the genome and is transmissible, and that these findings are not consistent with the hypothesis of SS14-lineage expansion purely due to macrolide resistance.” lines 290-292, “Such observations are not consistent with the hypothesis of an ancestrally resistant SS14 lineage driven to high frequency in the population due to a fitness advantage conferred by macrolide resistance, where we would expect to see expansion of a single resistant lineage.”

Could you provide a citation of the paper providing such hypothesis? The work of Arora et al. discusses omega SS14 cluster but the data clearly show that not all members of this cluster harbors macrolide resistant mutations and phylogenetic reconstruction of SS14 omega cluster shows small subclusters similar to sub-lineages described in this manuscript.

2/ The discovery of samples showing mixed character of 23S rRNA loci is surprising and controversial. Six samples analyzed previously by Pinto and colleagues are described to show 2 different alleles in 23S rRNA (wild type and A2058G) in this manuscript. How do you explain that the mixed character of these samples was missed by Pinto and colleagues in the original analysis? Moreover, this reviewer thinks that the different percentage of the resistant allele (24-94%) excludes the situation that one 23S rDNA locus contains sensitive allele and the other locus contains resistant allele, described as heterozygosity in phase by authors of this manuscript. In this scenario the reads would show roughly 50% of sensitive and 50% of resistant reads.

3/ Can you explain the mechanism of reversion of the resistant mutation to a wildtype stage suggested in sub-lineage 7? It is more probable that additional, compensatory mutations will appear instead of reversion of A2058G mutation. What is the fitness cost of A2058G mutation in TPA? The same situation occurs in sub-lineage 6 and 1A (only 1 wild type sample and the rest of the sub-lineage is macrolide resistant). How is it different from sub-lineage 7?

Reviewer #2:

Remarks to the Author:

This paper by Mathew A. Beale and colleagues reports the largest dataset for *Treponema pallidum*

whole genome sequences (WGSs) to date. The main claims of the paper are that lineages currently in circulation tend to be relatively recent (i.e. spread in the post-antibiotic era), but that resistance to azithromycin does not seem to play a role in the recent increase of syphilis prevalence, despite having emerged multiple times.

This is all reasonably interesting. Though, the paper is not always particularly clear and I found it difficult to read at times. I felt it did not help that the manuscript seemed to be largely framed as a rebuttal to an earlier paper by Arora et al. 2017.

T. pallidum is a remarkably difficult organism to get WGSs from. As such the authors should be commended for their effort that more than double the number of currently available *T. pallidum* genomes. On a slightly less positive note, the dataset is in fact not particularly large for current standards in microbial population genomics, with the addition of 73 new genomes combined with 49 previous publicly available ones.

More problematically, the dataset is also not particularly diverse geographically, with all new samples coming from the UK and the US. Possibly, even more worryingly, the new samples do not represent an ideal (random) sample as clinical manifestation is concerned. The UK samples were all collected from routinely diagnosed patients presenting with clinical evidence of syphilis at a single clinic in London. Whereas, the US samples were all collected in Seattle from individuals enrolled in a study of cerebrospinal fluid abnormalities in patients with syphilis.

This stratification by both fine-scale geography and clinical phenotype could lead to pseudo-replication as well as conflation between geography, genetic relatedness and phenotypes, which is likely to colour some of the key conclusions of the papers. The authors did not seem to consider the collinearity in geography, phenotype and genetics as a potential problem and made apparently no effort to disentangle possible confounders.

Below, I will comment specifically on the claims of the paper. Multiple independent acquisitions of the resistance mutation to azithromycin is not particularly surprising. In fact, I am not aware of any documented case where a resistance mutation arose only once in any bacterial species.

The inferred times to the Most Recent Common Ancestors (tMRCAs) are intriguing, in particular those for the SS14A and SS14B lineages that seem remarkably recent. Though, I have some reservations about the robustness of these findings given the sampling scheme, which may have missed out unsampled diversity.

Whilst I find it a priori unlikely that azithromycin may have driven the spread of *T. pallidum* over recent decades, I remain agnostic on this possibility. The authors claim this not to be the case, but without any real hard evidence being presented, beyond the observation that there are currently a large proportion of azithromycin-susceptible strains in circulation. One way to formally test for the role of azithromycin resistance in the recent spread of *T. pallidum* would be to contrast estimated basic reproduction rate (R_0) from the phylogeny between clades carrying, or not, the azithromycin resistance conferring SNPs. Though, I am afraid this is not really an option here, given the small number of independent clades and the highly biased sampling scheme.

I feel a bit sorry for sounding so negative. Despite its shortcomings, the bioinformatics/computational analyses seem globally sound and more than doubling the number of available genomes for a major pathogen remains a significant achievement. Also, somewhat ironically, and despite my criticisms, I feel the paper is far superior by many, if not most aspects to the Arora 2017 paper.

Minor points

I do not feel *T. pallidum* should be referred to as 'ancient pathogen'. The earliest historical records (supported by genomic analyses) date back to the Renaissance, which does not qualify as 'ancient'.

I personally would not refer to azythromycin as a second-line drug for *T. pallidum*. It may well be a suboptimal, if not misguided, drug, but to me, 'second-line' should be reserved to drugs deployed when 'first-line' compounds have been breached, which is not the case in syphilis.

I would suggest rewording the following statement "(meaning we would expect TPA genomes on average to accumulate one SNP every four years by natural drift)". What has been estimated in this paper is more akin to a mutation rate (rather than a substitution rate). As such, the driving force is mutation and not 'drift'. Fixation of mutation through selection or genetic drift only really matters when the samples in a phylogeny (the sequences at the tips of the tree) are lineages (e.g. species) rather than individual genomes.

Reviewer #3:

Remarks to the Author:

This is a very well designed study enrolling a fruitful collaboration of researchers from the syphilis field with researchers with expertise in microbial genomics. It essentially constitutes a logical continuation of two previous studies that studied the complete genome of *T. pallidum* directly from multiple clinical samples focusing either epidemiological or genomic issues.

It is very well written and the bioinformatics pipelines were correctly chosen and applied. The discussion flows very well and it is interesting to see that such a methodologically complex paper turns out to be a straightforward paper to read. I congratulate the authors for that.

I have two major concerns that are associated both with the relevance of the take-home message and with the lack of information regarding other antibiotic resistance markers that are likely more relevant for this kind of study.

Two major comments:

- The authors state that "Previous genomic analyses have suggested that one lineage of syphilis, called SS14, may have expanded recently, with most syphilis caused by this lineage, and that this expansion indicates emergence of a single pandemic azithromycin-resistant cluster". And the conclusion of the study now under review is that "macrolide resistance has independently evolved multiple times in *T. pallidum*". I think the reader will not be surprised with the take-home message because it makes all the sense that those would be the results that one could expect. In fact: i) the spread at "small-scale" of *T. pallidum* strains is done by sexual contact and does not depend on the genomic makeup of the bacterium; ii) the presence of antibiotic resistance markers is, with discrete exceptions (e.g., hitchhiking events during recombination) associated with selective pressure provided by antibiotic regimens; iii) macrolides are not a first option for treating syphilis but instead they are frequently provided to treat multiple other diseases, so the existence of sub-MIC of macrolides hardly yield "super-fitted" *T. pallidum* strains with higher transmission capabilities; and iv) the maintenance of such genetic markers is associated with fitness costs (this was well discussed by the authors in this paper). As such, my questions are: why would we expect a different scenario? Why not also expect the spread of genotypically macrolide-sensitive strains as well (as the authors concluded in this paper)? Why would one consider azithromycin-resistant strains more fitted than macrolide-sensitive strains so they would spread and the latter would not? So, my concern is that there is a lack of strong rationale that would make the authors' major conclusion a quite relevant take-home message;

- Considering the extent of this genomic survey focusing the chronological and geographical spread of an antibiotic resistance marker for macrolides and the fact that penicillin (and not the macrolides) is the treatment of choice for syphilis, I was very surprised to see that no results and discussion were presented regarding mutations in pbp or pbp-like proteins. In fact, Sun and colleagues (Sun et al. 2016. *Oncotarget*; 7(28): 42904-18) reported mutations in pbp2 (mrcA) and in other pbp genes for several clinical isolates, and Pinto et al (Pinto et al. 2016. *Nat Microbiology*. 2: 16190) reported a mutation also in mrcA for clinical isolates. The role of pbp genes is unquestionable and even the less evident "mrcA" may impact penicillin susceptibility as it includes a beta-lactamase active-site serine. On the other hand, phenotypic resistance to penicillin had not been reported in syphilis making these observations quite intriguing! As such, it is somehow hard to understand why the authors completely discarded such approach. It would turn this paper, in my opinion, into a much more clinically and epidemiologically relevant study. The enrolled teams are highly experienced and have all the tools to do that. I am sure it would make a great paper.

Other comments:

- Line 268: I presume that "consistent with other reports" the authors refer to refs 13 and 14, as these were the ones that observed this by WGS directly from clinical samples;

- Lines 278-282: I am not sure I agree with this "lucky speculation". The comparison of evolutionary patterns between *T. pallidum* and *C. trachomatis* is a real long shot. The singularity of the biphasic life-cycle of *C. trachomatis* makes it impossible to perform any comparison with *T. pallidum* besides pure speculation. *Rickettsia* and *Coxiella* are also intracellular bacteria. Are there any clues about the evolutionary rate that can be helpful for this speculation? And what about *M. tuberculosis*...? Are the authors comparing it with *C. trachomatis* just because the values luckily fit?

- Line 288: I believe that n=6 and n=3 refer to a specific sub-lineage and not to every sub-lineage...

- Line 289: Please clarify the sentence: "there were no sub-lineages representing an even mix of resistance genotypes";

- Lines 379-380: I presume this concentration refers to the eluted DNA (after Qiagen extraction) as no extrapolation of bacterial load can be made to the original samples because of the intermediate passages in rabbits. As such, why do the authors refer to borderline samples with high volume? Isn't the volume exactly the same for all eluted DNAs?

- Lines 383-384: Although this is not relevant for the paper, I am very curious about the used strategy. Why opting by adding human gDNA in order to reach the minimum advisable amount of >1ug/35ul (as suggested by the manufacturer)? There are other kits from the same manufacturer that allow using 200 ng and very recently, even 10 ng can be used. This avoids the unnecessary steps of "contaminating" the target DNA with human DNA, which can certainly interfere with the success.

- Line 386: Can the authors provide a little more details about the sonication process? It is known to be a delicate and crucial step but it is not clear if alternatives to the highly recommended Covaris are equally effective. This information could be useful for the scientific community within and outside the treponemal field.

- Line 389: For sure, the authors did not use the pure baits. The costs are prohibitive. They have certainly diluted them at least 1:10, right? Shouldn't this information be in the manuscript? It would be highly informative to the scientific community that never dares to use the baits due to their

prohibitive price if used non-diluted (between 6000€ and 10000 €, depending on the "concentration", for sets of 16 samples). This is the kind of information the scientific community would love to know.

Reviewers' comments:

Reviewer #1 (Remarks to the Author):

The manuscript “Genomic epidemiology of syphilis reveals independent emergence of macrolide resistance across multiple circulating lineages“ by Beale and colleagues describes the whole genome sequencing of 73 TPA samples from US and Europe and whole genome analysis of obtained data together with 48 publicly available TPA genomes.

This reviewer appreciates the amount of work done by the authors. However, although the sequencing of 73 TPA samples brings new data, conclusions and methodology are not novel:

We thank the reviewer for their comments and have addressed the specific comments below:

1) The sample enrichment using nucleic acid baits was described before (Arora et al. 2016, Pinto et al. 2016, Marks et al. 2018, Knauf et al. 2018)

The development of the sample enrichment approach was not the aim of this study, nor one of its findings. As such we had cited the Depledge (2011), Christiansen (2014), Pinto (2016),

Arora (2016) and Marks (2018) papers to acknowledge their work. Knauf et al's paper focussing on Yaws genomes in non-human primates had not been published when we submitted our manuscript to the journal. We have now added Knauf et al (2018) to the manuscript.

2) Emergence of macrolide resistance in different samples was described in a number of TPA typing papers (Chen et al. 2013, Grillova et al. 2014, Gallo Vaulet et al. 2017, Smajs et al. 2015).

We agree that these *Treponema pallidum* subsp. *pallidum* (TPA) typing papers previously showed that macrolide resistance patterns had been observed to differ by region (as demonstrated by Chen, 2013) or by molecular type/CDC type (Grimes 2012, Grillova, 2014 and Vaulet, 2017). Smajs (2015) attempted to compile global data on macrolide resistant syphilis from many papers, including those of Grillova and Chen, and we cited this in addition to the papers the reviewer mentions.

Importantly, Smajs et al (2015) make clear in their abstract, there is "...Scarce data regarding the genetics of macrolide-resistant mutations". To date, all substantive genetic correlation with macrolide resistance has been linked to the host population using comparatively low resolution molecular approaches such as the CDC typing scheme or multilocus sequence typing (MLST) schemes, not WGS as in this study. Molecular schemes such as MLST have limited ability in TPA to define accurate phylogenies and infer recent common ancestry, due to the low level of genomic variation evident in TPA. Therefore, they do not offer sufficient resolution to understand long-range evolutionary patterns that we describe here.

The novelty of our study is to use WGS to de-convolute these patterns of resistance across a broad set of samples from different geographic sites. None of the existing WGS studies have had sufficient breadth or depth of sampling to be able to do this; Arora et al sampled from multiple countries, but had relatively small numbers per location, whilst Sun et al and particularly Pinto et al sampled more deeply, but from single geographical locations. We combine data from all three papers with a larger number of new sequences, and by using WGS, for the first time we were able to provide an evolutionary framework that describes how individual sub-lineages differ from one another, and then start to place a time frame on observed events associated with resistance. Rather than simply seeing a list of molecular types and the proportion of isolates within that type that are resistant, the WGS phylogenetic framework is used to show how macrolide resistance has emerged and proliferated (we believe Figure 2B illustrates this point in particular).

To address these points we have now modified the manuscript to further emphasise that typing data (including adding additional references as suggested) had shown that there were differential patterns of resistance (Lines 99-102, 347-348) and discuss how these patterns of resistance fit in to an evolutionary framework for TPA (Lines 111-112).

The reviewer wants to stress that there is a number of statements throughout the paper, which are controversial and these include:

3) lines 39-43, "These findings demonstrate that macrolide resistance has independently evolved multiple times in *T. pallidum*, that once evolved it becomes fixed in the genome and is transmissible, and that these findings are not consistent with the hypothesis of SS14-lineage expansion purely due to macrolide resistance." lines 290-292, "Such observations are not consistent with the hypothesis of an ancestrally resistant SS14 lineage driven to high frequency in the population due to a fitness advantage conferred by macrolide resistance,

where we would expect to see expansion of a single resistant lineage.” Could you provide a citation of the paper providing such hypothesis?

We thank the reviewer for this point that this hypothesis is not explicitly stated in the literature, but rather discussed as a possibility in Nechvátal (2014) and further implied by the work of Arora (2016). In fact these studies informed the hypothesis we were testing, and so to clarify this issue we have edited the text to make this clear. Our data were not consistent with this hypothesis, but this underlines the novelty of our data. We have expanded our treatise of the comment made by Nechvátal (2014) to address this point and the one below (Lines 103-105, 108-110).

4) The work of Arora et al. discusses omega SS14 cluster but the data clearly show that not all members of this cluster harbors macrolide resistant mutations and phylogenetic reconstruction of SS14 omega cluster shows small subclusters similar to sub-lineages described in this manuscript.

It is true to say that Arora et al. shows a minimum spanning tree (Figure 2a) showing two SS14-lineage isolates (AR2 and CZ27) that are macrolide sensitive, but they neither commented on, nor made any conclusions relating to phylogenetic sub lineages of SS14 and the independent emergence of resistance, perhaps because they felt they had insufficient data to substantiate this.

5) The discovery of samples showing mixed character of 23S rRNA loci is surprising and controversial. Six samples analyzed previously by Pinto and colleagues are described to show 2 different alleles in 23S rRNA (wild type and A2058G) in this manuscript.

How do you explain that the mixed character of these samples was missed by Pinto and colleagues in the original analysis?

On the contrary, Pinto and colleagues showed that four samples had a mixed population of A2058G, describing three as having a “Nearly fixed mutation” and one A2058G “Probable emerging mutation” in their Figure 1. We included three of these four samples in our analysis (excluding PT_SIF1348 due to low overall coverage).

We identified an additional three genomes from the Pinto dataset with putative mixed 23S alleles. The discrepancy between our two analyses likely reflects the specifics of the methods used to infer mixed alleles, including mapping differences, coverage thresholds and frequency thresholds. For example, Pinto and colleagues considered minority variant sites to be mixed when the minority allele frequency was above 10%, whilst we used a cut off of 5%. In our analysis, samples PT_SIF1261 and PT_SIF1142 (described by Pinto et al as having a fixed 2058G allele) had 14.7% and 10.9% minority ‘A’ alleles. To address this question and clarify these points we have added to the discussion of this issue (lines 236-237).

6) Moreover, this reviewer thinks that the different percentage of the resistant allele (24-94%) excludes the situation that one 23S rDNA locus contains sensitive allele and the other locus contains resistant allele, described as heterozygosity in phase by authors of this manuscript. In this scenario the reads would show roughly 50% of sensitive and 50% of resistant reads.

We are inclined to agree with the reviewer that heterozygosity in phase is the least likely scenario (although it is conceivable that both heterozygosity in phase and mixed ‘superinfection’ might occur in high risk patients and sexual networks, confusing simple 50:50 frequencies). However, as heterozygosity between the two copies has been shown for *T. denticola* (Lee, 2002 – now addressed at lines 226-228), we considered the possibility that the same could be true for TPA, and given the caveats already described, we felt there

was insufficient evidence to conclusively rule this out and therefore left this an open question.

7) Can you explain the mechanism of reversion of the resistant mutation to a wildtype stage suggested in sub-lineage 7?

We anticipate the mechanism to be errors introduced through replication, followed by selection: in essence, by the same mechanism that the mutations appear in populations under drug selection.

8) It is more probable that additional, compensatory mutations will appear instead of reversion of A2058G mutation.

This of course would depend on the relative fitness cost of both events – a compensatory mutation would only be more likely to be selected for than a reversion mutation if the fitness burden were lower. We agree that compensatory mutations could indeed appear (and have expanded to text to discuss this at lines 358-361), and this might explain the stability of these resistance SNPs in the sub-lineages.

9) What is the fitness cost of A2058G mutation in TPA?

The fitness cost of the A2058G mutation in TPA is not known.

10) The same situation occurs in sub-lineage 6 and 1A (only 1 wild type sample and the rest of the sub-lineage is macrolide resistant). How is it different from sub-lineage 7?

The most parsimonious scenario for sublineage 7 is that resistance evolved in a node ancestral to this sublineage (such as the MRCA highlighted in Figure 2A as having strong posterior support). Within that lineage, the A2058G-negative sample is the most recent and shares a MRCA with a number of resistant samples, and if we allow that resistance had already evolved, this implies reversion to wild type. The alternative explanation for the resistance pattern we see would be that resistance evolved on at least three separate occasions in sublineage 7. Sublineage 6 is different because the A2058G-negative sample diverged from the rest of the sublineage before the MRCA in which it is likely the three resistant samples evolved resistance. Likewise, the evolution of the A2059G mutation in sublineage 1A likely occurred after the sensitive sample in that sublineage diverged from the rest. We have expanded the text at lines 275-280 to discuss this further.

Reviewer #2 (Remarks to the Author):

This paper by Mathew A. Beale and colleagues reports the largest dataset for *Treponema pallidum* whole genome sequences (WGSs) to date. The main claims of the paper are that lineages currently in circulation tend to be relatively recent (i.e. spread in the post-antibiotic era), but that resistance to azithromycin does not seem to play a role in the recent increase of syphilis prevalence, despite having emerged multiple times.

This is all reasonably interesting. Though, the paper is not always particularly clear and I found it difficult to read at times. I felt it did not help that the manuscript seemed to be largely framed as a rebuttal to an earlier paper by Arora et al. 2017.

T. pallidum is a remarkably difficult organism to get WGSs from. As such the authors should be commended for their effort that more than double the number of currently available *T. pallidum* genomes.

1) - On a slightly less positive note, the dataset is in fact not particularly large for current standards in microbial population genomics, with the addition of 73 new genomes combined with 49 previous publicly available ones.

We agree that in comparison to studies of many bacteria more amenable to culture our sample collection is smaller. However, as the reviewer acknowledges, sequencing of TPA is very challenging, as is obtaining sufficient temporally and geographically diverse samples with sufficient pathogen DNA to attempt sequence capture (e.g. swab samples are rarely kept in long term storage in the same way cultured bacterial isolates might be).

2) - More problematically, the dataset is also not particularly diverse geographically, with all new samples coming from the UK and the US.

With regards to geographical bias, it should be noted that many of the lineages formed by the Seattle sequences are heavily admixed with those from European patients (e.g. SS14 sub-lineages 2, 5, 7, 8), suggesting our population structure is not overly biased (we now highlight this at lines 195-200 and 320-326, as well as adding a new Supplementary Figure 2 to show the geospatial distribution of sub-lineages). In fact, because we have so many samples from Seattle this allows us to show the co-circulation of distinct macrolide sensitive and resistant samples in the same population, further supporting our conclusions. To take the example of the SS14-sublineages, whilst we agree that additional and more diverse sampling will likely reveal novel sub-lineages, this does not negate the finding of distinct sub-lineages within the SS14-lineage, nor of the striking association of these lineages with macrolide resistance genotyping.

3) - Possibly, even more worryingly, the new samples do not represent an ideal (random) sample as clinical manifestation is concerned. The UK samples were all collected from routinely diagnosed patients presenting with clinical evidence of syphilis at a single clinic in London. Whereas, the US samples were all collected in Seattle from individuals enrolled in a study of cerebrospinal fluid abnormalities in patients with syphilis.

Firstly, the Seattle samples were not universally from patients with CSF disease. Only 36% of Seattle genomes included here came from patients with evidence of CNS infection – we apologise for not making that clearer in the manuscript, and have now clarified this on lines 434-435.

Secondly, and as the reviewer correctly points out, the Seattle dataset, and the underlying MSM sexual network it reflects, could indeed potentially be confounded due to oversampling from a population enriched for central nervous system (CNS) disease, leading to pseudoreplication issues. At the same time, it is important to note that ~40% of persons with early syphilis have CNS invasion/involvement (Lukehart, 1988) – this largely goes unrecognized in most studies because clinicians don't routinely perform and take samples from lumbar punctures. Most studies that focus on randomly collected samples without testing for CNS invasion/involvement would therefore suffer from such a bias. Specific investigation of CNS invasion was beyond the scope of the current work, but unlike previous studies that did not look for evidence of CNS involvement for the reasons given above, we know that there could be CNS bias (36% of Seattle samples included were from patients with detectable CNS infection), yet we still see admixture between European and Seattle samples, suggesting this has no substantive effect on lineage structure or composition. This has also now been highlighted at lines 195-200 and 320-326 and in Supplementary Figure 2.

Regarding the London samples, they were indeed sampled from a single clinic that serves as a large sexual health centre in a major metropolis, meaning they are a random sample

from a diverse group of attending patients. Again, we see the London sequences obtained from this clinic admixed with sequence from other parts of Europe and the US.

This stratification by both fine-scale geography and clinical phenotype could lead to pseudo-replication as well as conflation between geography, genetic relatedness and phenotypes, which is likely to colour some of the key conclusions of the papers. The authors did not seem to consider the collinearity in geography, phenotype and genetics as a potential problem and made apparently no effort to disentangle possible confounders.

Please see our responses to this comment in the points above.

Below, I will comment specifically on the claims of the paper.

4) - Multiple independent acquisitions of the resistance mutation to azithromycin is not particularly surprising. In fact, I am not aware of any documented case where a resistance mutation arose only once in any bacterial species.

Smajs et al (2015) made clear in their abstract, there is “...Scarce data regarding the genetics of macrolide-resistant mutations”, with most prior data coming from comparatively lower resolution typing schemes, and this is the first study with sufficient genomes to allow deconvolution of the evolutionary relationships between macrolide resistant and sensitive strains.

We agree that the *de novo* emergence of antimicrobial resistance (including macrolides) has been well characterised in other bacteria both in the lab and in the clinic (Baker, 2018). A documented example of single lineage expansion would be in MRSA ST22-A2, where the expansion of this lineage across the UK was linked to the acquisition of *griA* and *gyrA* point mutations conferring fluoroquinolone resistance (Holden, 2013). There are also examples of the other extreme, such as *Chlamydia trachomatis*, where no circulating azithromycin resistance mutation can be detected in the clinical population.

Azithromycin is an important clinical intervention for many sexually transmitted organisms and it is likely that azithromycin has off target effects against TPA, particularly in high risk MSM networks where repeated- and co- infection are more likely. More broadly, azithromycin is critical to eradication efforts for *T. pertenuis* (Yaws) and so it is important to understand how resistance can emerge in these closely related subspecies. Given the reported shortages of BPG, azithromycin could become more important for TPA and, as discussed earlier, macrolide selection pressure could provide an explanation for the apparent dominance of the SS14-lineage in currently sequenced datasets. Therefore, we felt it was an important hypothesis to test.

5) - The inferred times to the Most Recent Common Ancestors (tMRCAs) are intriguing, in particular those for the SS14A and SS14B lineages that seem remarkably recent. Though, I have some reservations about the robustness of these findings given the sampling scheme, which may have missed out unsampled diversity.

It is true that we cannot rule out that there is unsampled diversity. However, from what we and others have observed so far, TPA diversity *per se* is remarkably low, as it also is in other *Treponema pallidum* subspecies (e.g. Marks, 2018). To make our date estimates as robust as possible we excluded samples for which the provenance was not clear (for example, Arora et al describe using a range of years for the date of two Nichols genomes, but the only date shown in their ‘Supplementary Table 1’ is 1912, despite presumably around 100 years of passage). We also focussed on samples that we could be certain were recently derived from clinical samples and which had not undergone extensive passage in rabbits in an attempt to mitigate the potential biases. Our root-to-tip plot (Supplementary Figure 1) shows

that the confidence intervals broaden as we move further backwards in time – thus they are narrower during the 1990s when we date the origins of the key ancestral nodes for our SS14 sub-lineages than our dating for the origins of SS14-A/B. It is important to note that the molecular rate represents a mean from a distribution of rates, and this is reflected in the confidence intervals we show in Figure 3 for the TMRCA. We should have added 95% HPD for this rate and have now done so on line 173.

Notwithstanding the above, and since our interest was in recent evolutionary events, we confined our analysis to the parts of the tree where the confidence intervals were narrow, and did not attempt to make meaningful inferences from the more ancestrally distant nodes such as the divergence of Nichols and SS14, or SS14A and SS14B.

6) - Whilst I find it *a priori* unlikely that azithromycin may have driven the spread of *T. pallidum* over recent decades, I remain agnostic on this possibility. The authors claim this not to be the case, but without any real hard evidence being presented, beyond the observation that there are currently a large proportion of azithromycin-susceptible strains in circulation. One way to formally test for the role of azithromycin resistance in the recent spread of *T. pallidum* would be to contrast estimated basic reproduction rate (R_0) from the phylogeny between clades carrying, or not, the azithromycin resistance conferring SNPs. Though, I am afraid this is not really an option here, given the small number of independent clades and the highly biased sampling scheme.

Our data do indeed show that there has been an expansion of both sensitive and resistance lineages in different, clinically relevant geographic sites and also within the same population from Seattle. If there were a strong and consistent selective pressure imposed by azithromycin, we would expect to see expansion of resistant lineages and decline of sensitive lineages. This supports our statement that azithromycin-resistance *per se* has not been critical in the spread of *T. pallidum* over recent decades. Thank you for the suggestion to contrast R_0 between lineages; this will be possible in the future as this field develops and genomic investigations of TPA become more commonplace.

I feel a bit sorry for sounding so negative. Despite its shortcomings, the bioinformatics/computational analyses seem globally sound and more than doubling the number of available genomes for a major pathogen remains a significant achievement. Also, somewhat ironically, and despite my criticisms, I feel the paper is far superior by many, if not most aspects to the Arora 2017 paper.

We thank the reviewer for their kind feedback.

Minor points

I do not feel *T. pallidum* should be referred to as 'ancient pathogen'. The earliest historical records (supported by genomic analyses) date back to the Renaissance, which does not qualify as 'ancient'.

We agree, and have altered line 53 in the text to state that syphilis is 'centuries-old'.

I personally would not refer to azithromycin as a second-line drug for *T. pallidum*. It may well be a suboptimal, if not misguided, drug, but to me, 'second-line' should be reserved to drugs deployed when 'first-line' compounds have been breached, which is not the case in syphilis.

We have altered the text at line 402 to reduce the possibility for misunderstanding that azithromycin can be considered 'second line'.

I would suggest rewording the following statement "(meaning we would expect TPA

genomes on average to accumulate one SNP every four years by natural drift)". What has been estimated in this paper is more akin to a mutation rate (rather than a substitution rate). As such, the driving force is mutation and not 'drift'. Fixation of mutation through selection or genetic drift only really matters when the samples in a phylogeny (the sequences at the tips of the tree) are lineages (e.g. species) rather than individual genomes.

We have rephrased **line 175** to remove the term 'natural drift'.

Reviewer #3 (Remarks to the Author):

This is a very well designed study enrolling a fruitful collaboration of researchers from the syphilis field with researchers with expertise in microbial genomics. It essentially constitutes a logical continuation of two previous studies that studied the complete genome of *T. pallidum* directly from multiple clinical samples focusing either epidemiological or genomic issues.

It is very well written and the bioinformatics pipelines were correctly chosen and applied. The discussion flows very well and it is interesting to see that such a methodologically complex paper turns out to be a straightforward paper to read. I congratulate the authors for that.

We thank the reviewer for the positive comments.

I have two major concerns that are associated both with the relevance of the take-home message and with the lack of information regarding other antibiotic resistance markers that are likely more relevant for this kind of study.

Two major comments:

- The authors state that "Previous genomic analyses have suggested that one lineage of syphilis, called SS14, may have expanded recently, with most syphilis caused by this lineage, and that this expansion indicates emergence of a single pandemic azithromycin-resistant cluster". And the conclusion of the study now under review is that "macrolide resistance has independently evolved multiple times in *T. pallidum*". I think the reader will not be surprised with the take-home message because it makes all the sense that those would be the results that one could expect. In fact: i) the spread at "small-scale" of *T. pallidum* strains is done by sexual contact and does not depend on the genomic makeup of the bacterium; ii) the presence of antibiotic resistance markers is, with discrete exceptions (e.g., hitchhiking events during recombination) associated with selective pressure provided by antibiotic regimens; iii) macrolides are not a first option for treating syphilis but instead they are frequently provided to treat multiple other diseases, so the existence of sub-MIC of macrolides hardly yield "super-fitted" *T. pallidum* strains with higher transmission capabilities; and iv) the maintenance of such genetic markers is associated with fitness costs (this was well discussed by the authors in this paper).

1) As such, my questions are: why would we expect a different scenario?

The apparent global dominance of SS14 was discussed in Nechvátal (2014): *"In contrast to world-wide predominance of SS14-like strains, more than a half of the reference laboratory strains belong to Nichols-like group... ...A possible explanation of this fact is the macrolide resistance of the TPA SS14 strain"*. Furthermore, Arora et al (2016) describe a *"pandemic azithromycin-resistant cluster"* that *"diversified from a common ancestor in the mid-twentieth century subsequent to the discovery of antibiotics"*.

There are examples of single bacterial lineages being driven to dominance due to selection of antimicrobial resistance alleles, both through acquisition by horizontal gene transfer (HGT) and also through point mutations (Baker, 2018), for example: the expansion of MRSA

ST22-A2 across the UK after acquisition of two point mutations generating amino acid substitutions in GrlA and GyrA, conferring resistance to fluoroquinolones (Holden, 2013). In the light of prior literature, we believe it was reasonable to test such a scenario, and our data support the null hypothesis of multiple independent events leading to resistance.

2) Why not also expect the spread of genotypically macrolide-sensitive strains as well (as the authors concluded in this paper)?

Many thanks, yes we agree that would be the competing scenario to the one we tested. This is mentioned in our conclusion.

3) Why would one consider azithromycin-resistant strains more fitted than macrolide-sensitive strains so they would spread and the latter would not? So, my concern is that there is a lack of strong rationale that would make the authors' major conclusion a quite relevant take-home message;

Sorry if we were not clear. The implication from our study is that resistance may have been selected for in the past, perhaps during the 1990s and early 2000s as a result of azithromycin prophylaxis in HIV-infected persons, leading to the emergence of multiple resistant lineages. There is no strong evidence for a continued selective pressure by azithromycin on TPA in our data, as evidenced by the subsequent expansion of both sensitive and resistant lineages.

4) Considering the extent of this genomic survey focusing the chronological and geographical spread of an antibiotic resistance marker for macrolides and the fact that penicillin (and not the macrolides) is the treatment of choice for syphilis, I was very surprised to see that no results and discussion were presented regarding mutations in pbp or pbp-like proteins. In fact, Sun and colleagues (Sun et al. 2016. *Oncotarget*; 7(28): 42904-18) reported mutations in pbp2 (*mrcA*) and in other pbp genes for several clinical isolates, and Pinto et al (Pinto et al. 2016. *Nat Microbiology*. 2: 16190) reported a mutation also in *mrcA* for clinical isolates. The role of pbp genes is unquestionable and even the less evident "mrcA" may impact penicillin susceptibility as it includes a beta-lactamase active-site serine. On the other hand, phenotypic resistance to penicillin had not been reported in syphilis making these observations quite intriguing! As such, it is somehow hard to understand why the authors completely discarded such approach. It would turn this paper, in my opinion, into a much more clinically and epidemiologically relevant study. The enrolled teams are highly experience and have all the tools to do that. I am sure it would make a great paper.

Thank you for this suggestion, we have now investigated this. We selected the three protein coding genes with predicted penicillin binding activity analysed by Sun et al and Pinto et al (*pbp1*, *pbp2*, *mrcA*), as well as the Tp47 protein described by Cha et al (2004) as having novel beta-lactamase activity. These genes were screened across the dataset for the presence of non-synonymous SNPs, as well as using SIFT to make predictions about the functional impact of such amino acid changes.

Our analysis showed that there are a number of SNPs that are highly conserved across TPA lineages and sub-lineages (e.g. the *mrcA* G708S variant described by Pinto et al). In spite of *in silico* predictions of functional effect, there is no evidence of actual clinical treatment failure, and it is highly unlikely that such widely distributed SNPs truly impact on penicillin resistance to a significant degree *in vivo*. However, we did find a small number of isolated SNPs that might be of interest. Of particular note, we observe *mrcA* amino acid changes at position A506, both the A506V described by Sun and colleagues, as well as A506T which occurs both in a group of US SS14 samples, as well as in the clinically-derived Nichols genomes. That this amino acid site appears to have been targeted on 3 separate occasions by homoplastic mutations is highly suggestive of selection. However, as the reviewer points

out, there is no reported phenotypic penicillin resistance, thus making validation of any such SNPs unfeasible.

To address this point we have now added a section on penicillin associated gene variants, with expansion of the methods (lines 546-553) and results (lines 289-312), a small addition to the discussion (lines 411), and a supplementary figure (Supplementary Figure 3) to describe this information.

Other comments:

5) - Line 268: I presume that “consistent with other reports” the authors refer to refs 13 and 14, as these were the ones that observed this by WGS directly from clinical samples;

We cited Arora et al, as this was the only study that attempted to perform a ‘global’ analysis of whole TPA genomes; Pinto et al and Sun et al confined their sampling to a single country or city (with a few global reference sequences for context), and all of their novel sequences were from SS14-lineage.

6) - Lines 278-282: I am not sure I agree with this “lucky speculation”. The comparison of evolutionary patterns between *T. pallidum* and *C. trachomatis* is a real long shot. The singularity of the biphasic life-cycle of *C. trachomatis* makes it impossible to perform any comparison with *T. pallidum* besides pure speculation. *Rickettsia* and *Coxiella* are also intracellular bacteria. Are there any clues about the evolutionary rate that can be helpful for this speculation? And what about *M. tuberculosis*...? Are the authors comparing it with *C. trachomatis* just because the values luckily fit?

We agree, although to our knowledge there are no large WGS-based studies of the molecular rates for *Coxiella* or *Rickettsia*. However, to address this point we have expanded the text and added details of the mutation rate for *M. tuberculosis* (doubling time 15-20hrs, 1×10^{-7} SNPs/site/year) as an additional comparison (Lines 334-337).

7) - Line 288: I believe that n=6 and n=3 refer to a specific sub-lineage and not to every sub-lineage...

Thank you for this. It referred to 6 genotypically resistant sublineages and 3 genotypically sensitive sublineages. We have clarified this at lines 345-346

8) - Line 289: Please clarify the sentence: “there were no sub-lineages representing an even mix of resistance genotypes”;

We agree this was confusing and have removed the statement from the text at lines 345-347.

9) - Lines 379-380: I presume this concentration refers to the eluted DNA (after Qiagen extraction) as no extrapolation of bacterial load can be made to the original samples because of the intermediate passages in rabbits. As such, why do the authors refer to borderline samples with high volume? Isn't the volume exactly the same for all eluted DNAs?

Yes, the sample volume at extraction was the same, but because many of the precious samples we sequenced had previously been used for other molecular studies, we were working with ‘residual DNA extracts’; thus the sample volume we received was variable. We have clarified this in the methods on line 442-443.

10) - Lines 383-384: Although this is not relevant for the paper, I am very curious about the used strategy. Why opting by adding human gDNA in order to reach the minimum advisable amount of >1ug/35ul (as suggested by the manufacturer)? There are other kits from the same manufacturer that allow using 200 ng and very recently, even 10 ng can be used. This avoids the unnecessary steps of “contaminating” the target DNA with human DNA, which can certainly interfere with the success.

The bulking of low concentration DNA/cDNA samples with commercial DNA is a standardised strategy for target enrichment sequencing, and was described in both Dan Depledge’s original 2011 paper on enrichment of viral genomes and in Mette Christiansen’s 2014 Chlamydia paper (plus many others including our own). In most cases, samples are already contaminated with substantial quantities of DNA from the host. Counter-intuitively, adding more host DNA can actually help rather than hinder success, reducing the loss of the low concentration target DNA during the sequence tag DNA ligation stages in Illumina library construction, in favour of the proportionally higher loss of the added bulk DNA. With a highly specific pull down array most of this bulk DNA is then removed during the hybridisation enrichment process, and computational removal of residual reads is relatively trivial (particularly given we would be doing it anyway for residual host reads). This has proven highly successful for us and so we applied the same approach to Treponemal sequencing.

11) - Line 386: Can the authors provide a little more details about the sonication process? It is known to be a delicate and crucial step but it is not clear if alternatives to the highly recommended Covaris are equally effective. This information could be useful for the scientific community within and outside the treponemal field.

We used a Covaris LE220 ultrasonicator (average size distribution ~150bp) – this detail has been added to the text at line 455.

12) - Line 389: For sure, the authors did not use the pure baits. The costs are prohibitive. They have certainly diluted them at least 1:10, right? Shouldn’t this information be in the manuscript? It would be highly informative to the scientific community that never dares to use the baits due to their prohibitive price if used non-diluted (between 6000€ and 10000 €, depending on the “concentration”, for sets of 16 samples). This is the kind of information the scientific community would love to know.

In contrast to many other approaches for target enrichment (e.g. Hodges (2009), Depledge et al (2011), Christiansen et al (2014), Pinto et al (2016) and Arora et al (2016)) which used discrete ‘per-sample’ pulldowns analogous to the SureSelect XT1 approach, our method used here and in Marks et al (2018) is more analogous to the SureSelect XT2 protocol, and performs initial library preparation, adaptor ligation and sample indexing/barcoding, before creating equimolar pools of total DNA from a number of samples (in the case of this study, 20 samples). Only after samples have been pooled together is a hybridisation pulldown performed. This allows us to use neat undiluted baits, but multiplex 20 samples per hybridisation reaction, thus reducing library prep as well as bait costs. To address this point we have now added extra detail to our methods section at lines 454-467.

Reviewers' Comments:

Reviewer #1:

Remarks to the Author:

The resubmitted manuscript „Genomic epidemiology of syphilis reveals independent emergence of macrolide resistance across multiple circulating lineages“ by Beale and colleagues shows significant improvements in clarity of presented data.

The reviewer's comments were somehow addressed, but ironically the authors' response opens new questions because changes made in the manuscript revealed missing analysis for certain points made by authors and a significant discrepancy is presented regarding the macrolide resistance mutation data.

Major points:

The following numbering of pages is based on the newly submitted manuscript (not the modified version of the original manuscript).

1/ Regarding number 6 of author's response to reviewer 1, mixed 'superinfection' in high risk patients and sexual networks as described by the authors could be probably traced in the data of WGS with coverage as high as presented data. Given that authors speculate that a patient could be infected by a mixture of two (or more) different isolates, analysis of data could reveal the positions of genome in which these two (or more) strains differ. Did you detect any genetic variability in the data that could suggest this phenomenon? This is an important point and should be mentioned in the paper.

2/ Regarding number 10 of author's response to reviewer 1, lines 259-261, "We also observe an instance of possible reversion from resistant to wildtype allele in sub-lineage 7, with a wildtype sample (SW6) descending from multiple putatively resistant MRCA nodes;" Sample SW6 presented here as the only macrolide sensitive sample within sublineage 7, being discussed here as a highly speculative "possible reverse mutation" is in fact presented as macrolide resistant (A2058G) in Figure 2 and Supplementary Table 8 of Arora paper published in 2016. Can you explain why your analysis of sequencing data showed this sample as macrolide sensitive?

3/ Regarding number 8 of author's response to reviewer 1, lines 337-341, "That the variants appear stable within lineages, with only a single instance in the phylogeny that might represent reversion to a wildtype state, suggests that either there is no strong fitness cost associated with possessing these macrolide resistance mutations or that novel compensatory variants co-evolve to mitigate any fitness cost, enabling variant stability; further sampling would be necessary in the future to address this." Did you observe any genetic changes that are linked to the presence of macrolide resistance mutation (thus possible novel compensatory variants)? It is probable that sampling of this study is sufficient to suggest such a thing.

4/ Lines 97-98: "These findings have implications for our understanding of the increasing incidence of syphilis..." How does the main conclusion of the paper (that macrolide resistance evolves in different sublineages independently) contribute to our understanding of the increasing incidence of syphilis?

Minor points (could help to orient in the Figures better):

Figure 2. Inclusion of sample names is probably not feasible, but country of origin column could help to compare info in this Figure with Figure 1.

Supplementary Figure 3. Column with colours indicating lineage would be helpful.

Reviewer #2:

Remarks to the Author:

I wish to thank the authors for having engaged with my criticisms. I feel they have satisfactorily addressed the points I raised that that could be dealt with, given the dataset. To me, the revision is significantly improved and I found the new section on putative penicillin resistant mutations very interesting.

Francois Balloux

Reviewer #3:

Remarks to the Author:

As I stated in the previous review round, I still have some doubts regarding the validity of the hypothesis raised by the authors, which somehow questions the novelty of the take-home message. In my opinion (and it looks like it is also the opinion of other reviewers) the results are not surprising. Nevertheless, I reaffirm that the methodological approach is excellent as well as the bioinformatics analyses. And this is even more relevant in such a tricky pathogen (likely the most complicated I have worked with) with such a small number of sequenced genomes. I congratulate the authors for that. It was also very interesting to see that the authors deeply analyzed some of the previous published studies that used similar approaches and were thus comfortable in comparing the highly complex results. This comparison was essential to validate their data and for the flow of this paper. I truly appreciate that the authors now included several experimental details that were lacking. And I also have no doubts that the "rationale" of the paper is now better explained with the inclusion of several paragraphs throughout the different sections.

João Paulo Gomes

REVIEWERS' COMMENTS:

Reviewer #1 (Remarks to the Author):

The resubmitted manuscript „Genomic epidemiology of syphilis reveals independent emergence of macrolide resistance across multiple circulating lineages“ by Beale and colleagues shows significant improvements in clarity of presented data.

The reviewer's comments were somehow addressed, but ironically the authors' response opens new questions because changes made in the manuscript revealed missing analysis for certain points made by authors and a significant discrepancy is presented regarding the macrolide resistance mutation data.

Major points:

The following numbering of pages is based on the newly submitted manuscript (not the modified version of the original manuscript).

1/ Regarding number 6 of author's response to reviewer 1, mixed 'superinfection' in high risk patients and sexual networks as described by the authors could be probably traced in the data of WGS with coverage as high as presented data. Given that authors speculate that a patient could be infected by a mixture of two (or more) different isolates, analysis of data could **reveal the positions of genome in which these two (or more) strains differ**. Did you detect any genetic variability in the data that could suggest this phenomenon? This is an important point and should be mentioned in the paper.

We thank the reviewer for this comment, but wish to again clarify that the original finding of mixed alleles at 23S sites was not ours, but was described by Pinto and colleagues – in our analysis we primarily used the same data and replicated the findings of Pinto. The reviewer raises an interesting point that is worth exploring further. However, we need to stress that analysis of within-host diversity, as well as discrimination between within-host evolution and superinfection from direct sequencing data is in no way a trivial task, as evidenced by the large body of literature on the subject in the virology field. Whilst there are highly developmental methods we could attempt, this would require a substantial change in direction for the manuscript as it represents an entirely different question to the one we attempted to address. Furthermore, our dataset was not optimally designed to address questions around within host diversity: we had no control over the sampling and sequencing of the publicly available genomes, and only 8 of our new genomes were sequenced directly from patients, with the remainder subjected to limited passage in rabbits – an act likely to alter the infection bottle neck and thus the within-host dynamics of infection. The reviewer can be reassured that this is something we will look to address in future studies with more appropriately designed sample groups and sequencing strategies.

2/ Regarding number 10 of author's response to reviewer 1, lines 259-261, “We also observe an instance of possible reversion from resistant to wildtype allele in sub-lineage 7, with a wildtype sample (SW6) descending from multiple putatively resistant MRCA nodes;” Sample SW6 presented here as the only macrolide sensitive sample within sublineage 7, being discussed here as a highly speculative “possible reverse mutation” is in fact presented as macrolide resistant (A2058G) in Figure 2 and Supplementary Table 8 of Arora paper published in 2016. Can you explain why your analysis of sequencing data showed this sample as macrolide sensitive?

We thank the reviewer for bringing this to our attention. We have thoroughly investigated this discrepancy, and have identified two issues. Firstly, a small number of the downloaded publicly

available read sets (including those from SW6) were incorrectly processed by our automated metagenomic binning pipeline, meaning that contaminating (non-Treponema) reads were not adequately removed prior to analysis. This relatively minor issue was compounded by a second observation: there is an extremely abnormal coverage distribution approximately 100 bp upstream of the A2058 locus in 23S for the majority of sequence reads published by Arora and Pinto, as demonstrated in the annotated Artemis screenshot below (using reads from sample SW6, published as accession SRR3268732).

In our coverage graph above (bottom panel), the mean genome coverage for the A2058G site is actually around 150x, but this is not visible because there is a region immediately upstream of the 2058/2059 locus with coverage of over 75,000x mapped reads. Further investigation of this small region of increased coverage shows that it is extremely conserved across the bacterial kingdom, matching at 100% kmer identity to over 320,000 bacterial datasets in the ENA.

This is a phenomenon we have observed previously with Chlamydia sequencing (Hadfield et al, 2017), and occurs when baits used for sequence capture over-enrich for a genomic region common to many bacterial species present in the original sample. Interestingly, we do not observe this coverage spike in either non-enriched metagenomics sequences from Sun et al (2016), or in enriched data from our own study, and we speculate that there must be a specific 120-mer probe common to the Arora and Pinto probe sets. These reads are identical to the Treponema 23s rRNA sequence over the small highly conserved region we describe (alongside the site where SNPs associated with macrolide resistance occur) but differ elsewhere and in the mapping of the paired mate reads. To confirm this, since these studies used pair-end sequencing, we looked for the mate pairs for reads mapping to this region. For Treponema they should lie within the range of the Illumina library insert sizes. For other bacteria the mate pairs should not map, since outside of this conserved region the sequence of 23s rRNA is not so well conserved across bacteria.

For our original analysis of macrolide resistance we used ARIBA, which performs localised assembly and mapping to infer alleles and summarises the results – our validation with our own samples had shown no issues. In the particular case of sample SW6 from the Arora study, taxonomic classification of the raw reads (available as SRR3268732) with Kraken/Braken shows that the sample is heavily contaminated with other species, with around 29% of sequencing reads classified as *Streptococcus disgalactiae*. Detailed inspection of the assemblies generated by ARIBA for the contaminated SW6 sample shows that the contaminating coverage spikes we describe can lead to miss assembly, since the reads at this point contain a mixture of matching and mismatching mate pairs in the paired reads. The problem with ARIBA was only partially solved by correcting the failed *Treponema*-read binning step. This resulted in the high coverage of contaminating orphan reads being included in the original assembly and mapping and explains the discordance with the original publication: reanalysis shows that SW6 does indeed carry the A2058G mutation.

Unfortunately we missed this in our original analysis, we thank the reviewer for bringing this to our attention and to address this point we have:

- 1) Reanalysed the 23S and PBP gene variants for the entire dataset using a variant on the highly stringent competitive mapping strategy previously described in Hadfield et al (Genome Res, 2017) and illustrated below using a figure from that paper.

[redacted]

- 2) Repeated the *Treponema* read binning for the affected samples (ensuring that it worked as expected). The entire dataset was remapped to the reference genome, all multiple sequence alignments were rebuilt, and all phylogenies reconstructed, including repeating the BEAST analyses.

This reanalysis has had the following effects:

1. Sample SW6 is now labelled as positive for the A2058G allele – as this was the only plausible evidence for reversion, we have amended the text that described this event as appropriate on line 290 and at line 369 in the tracked changed document.
2. In cases where the competitive mapping showed resistance was very likely (and the mixture was due to clear contaminants), some alleles in 23S previously described as ‘mixed’ are now labelled as ‘resistant’ (no mixed samples have become ‘negative’. Rather than remove the original analysis, we have added a paragraph to the Results at lines 233-256 in the tracked changes document describing this reanalysis and it’s importance. We have also added a section to the Methods on lines 582-596 describing the competitive remapping approach.
3. For other samples, competitive mapping still showed evidence of mixed alleles, but for safety, we have labelled 23S alleles for some samples previously described as ‘mixed’ as

‘uncertain’. We have applied the same approach to the penicillin binding protein variants, as well as ensuring that sites where there is insufficient read coverage to definitively call the allele either way (negative OR positive) are also described as ‘uncertain’. A small number of PBP variants affecting only single samples appear to have been spurious – these have been removed from the supplementary figure – the distribution of all alleles affecting multiple samples remain effectively the same.

4. Recalling WGS variants after repeating the Treponema binning lead to some minor differences, as is expected when read counts are changed (particularly for low coverage sequencing). Rebuilding the recombination-masked SNP-only alignment with these data lead to a reduction in the number of parsimony informative sites from 284 to 276. The result of this is that the phylogenies show minor topological changes in areas of the tree that were previously poorly supported (low bootstrap or posterior), but the fundamental clustering of strains remains the same.
5. The molecular rate we observe from the BEAST tree has changed from 2.28×10^{-7} to 1.78×10^{-7} sites/year (a change from 1 SNP/4 years to 1 SNP/5 years. This has no substantive effect on the origins of the sub-lineages analysed in our study, but does affect the ancestral date of separation of SS14-lineage from Nichols-lineage (which already had broad confidence intervals, and was not the subject of our study).

Importantly, although these changes have slightly altered the topology of the tree and changed the dating, they did not affect the overall findings and conclusion of the paper: that macrolide resistance alleles are strongly associated with phylogenetic sub-lineage. Once again, we thank the reviewer for bringing this to our attention, and have acknowledged this important contribution in the manuscript acknowledgements.

3/ Regarding number 8 of author’s response to reviewer 1, lines 337-341, “That the variants appear stable within lineages, with only a single instance in the phylogeny that might represent reversion to a wildtype state, suggests that either there is no strong fitness cost associated with possessing these macrolide resistance mutations or that novel compensatory variants co-evolve to mitigate any fitness cost, enabling variant stability; further sampling would be necessary in the future to address this.” Did you observe any genetic changes that are linked to the presence of macrolide resistance mutation (thus possible novel compensatory variants)? It is probable that sampling of this study is sufficient to suggest such a thing.

There are no variants within the 23S gene itself that co-segregate with the 2058/2059 resistance alleles. Looking beyond 23S, we made clear on line 341 of the revised manuscript that “*further sampling would be necessary in the future to address this*” question. This is because any test of association would have to account for population structure – variants associated with a particular resistant sublineage could either be compensatory or merely coincidental along the branches leading to each sublineage. We attempted to probe the dataset for compensatory variants during our early analysis for the paper using TreeWAS (Collins and Didelot, PLoS Comp Bio 2018), but unsurprisingly this showed no signal as our study is not powered for this type of analysis.

We also note that with the correction to the A2058G status of sample SW6, there is no longer evidence for reversion in our phylogeny, further supporting our point about variant stability. We have therefore rephrased the quoted sentence from line XXX: “*That the variants appear stable within lineages, with no examples in ~~only a single instance in the phylogeny~~ that might represent reversion to a wildtype state...*”

4/ Lines 97-98: “These findings have implications for our understanding of the increasing incidence of syphilis...” How does the main conclusion of the paper (that macrolide resistance evolves in different sublineages independently) contribute to our understanding of the increasing incidence of syphilis?

In this study, we tested whether the expansion of the SS14-lineage (which appears to be the dominant clone in the currently published sequencing datasets, as well as our own) was driven by macrolide selection pressure, as postulated by Nechvátal (2014), and suggested by the repeated mention of a “pandemic azithromycin-resistant cluster” that “diversified from a common ancestor in the mid-twentieth century subsequent to the discovery of antibiotics” in Arora (2016). Our findings challenge this, showing that single clone expansion of a macrolide resistant lineage has not occurred (it was separate, independent sublineages, some of which are resistant, others of which are not), thus implying that the current SS14-lineage expansion is not driven by macrolide selection. We have amended the text at lines 98-100 to address this point:

*“These findings challenge the idea that expansion of the SS14 lineage has been driven by macrolide resistance, and have implications for the potential of the WHO Yaws eradication campaign to drive further development of macrolide resistance in both TPA and in the closely related *Treponema pallidum* subspecies *pertenue* (TPP)”.*

Minor points (could help to orient in the Figures better):

Figure 2. Inclusion of sample names is probably not feasible, but country of origin column could help to compare info in this Figure with Figure 1.

Thank you for this – it was something we considered ourselves, but we ultimately decided against this approach so as to not over complicate or dilute the impact of Figure 2 – the key message of Figure 2 is the macrolide resistance genotyping correlating with the lineages. However, we agree that it would be helpful for the reader to be able to check lineages by country, and have now added the suggested plot as a new supplementary figure (Supplementary Figure 2), and of course this information is also contained in Supplementary Table 1.

Supplementary Figure 3. Column with colours indicating lineage would be helpful.

We have made this change as suggested to what is now Supplementary Figure 4, as well as changing the colour scheme for the SNPs to be more consistent with the other figures.

Reviewer #2 (Remarks to the Author):

I wish to thank the authors for having engaged with my criticisms. I feel they have satisfactorily addressed the points I raised that that could be dealt with, given the dataset. To me, the revision is significantly improved and I found the new section on putative penicillin resistant mutations very interesting.

Francois Balloux

We thank the reviewer for their helpful feedback.

Reviewer #3 (Remarks to the Author):

As I stated in the previous review round, I still have some doubts regarding the validity of the hypothesis raised by the authors, which somehow questions the novelty of the take-home message. In my opinion (and it looks like it is also the opinion of other reviewers) the results are not surprising. Nevertheless, I reaffirm that the methodological approach is excellent as well as the bioinformatics analyses. And this is even more relevant in such a tricky pathogen (likely the most complicated I have worked with) with such a small number of sequenced genomes. I congratulate the authors for that. It was also very interesting to see that the authors deeply analyzed some of the previous published studies that used similar approaches and were thus comfortable in comparing the highly complex results. This comparison was essential to validate their data and for the flow of this paper. I truly appreciate that the authors now included several experimental details that were lacking. And I also have no doubts that the "rationale" of the paper is now better explained with the inclusion of several paragraphs throughout the different sections.

João Paulo Gomes

We thank the reviewer for their helpful feedback.

Reviewers' Comments:

Reviewer #3:

Remarks to the Author:

At this final stage, I was asked to deeply analyze the modifications performed in the manuscript, on behalf of the comments of another reviewer (reviewer 1). As a result of those comments essentially regarding potential bias associated with SNPs conferring macrolide resistance, the authors repeated the whole analysis and used additional bioinformatics tools to ensure the degree of certainty of the final data. Their final results were basically the same, with only minor changes that did not affect the major conclusions. I must say that only very experienced groups as this one would be capable of such "fine-tune" analysis. Sometimes doing many extra analysis may be problematic because we always find very tiny details for which there is no reasonable explanation. The WGS data is very "rich" on these annoying issues.

As a result of this final re-analysis, they corrected some sentences and added others to explain the additional procedure and the reason for doing it. Finally, the explanation regarding their inability to study the within-host evolution is correct as most of the new genomes were passaged in rabbits, which certainly creates sub-population bottlenecks and hampers such analysis. Nevertheless, although this is not very relevant for the present study, the authors were, in my opinion, too cautious when they say that it is not trivial to distinguish between mixed infections and different alleles of a single strain for *T. pallidum*. Whereas this is completely valid for a prevalent pathogen, that is clearly not the case for the highly infrequent *T. pallidum*. So, although we can never be sure when we face different alleles, the probability of a scenario of within-host evolution is tremendously higher than the one of a mixed infection.

Overall, I believe the authors have elegantly addressed the concerns raised by reviewer 1.

João Paulo Gomes